# Cosmic-Ray Extremely Distributed Observatory

**Piotr Homola** [1,*], **Dmitriy Beznosko** [2], **Gopal Bhatta** [3], **Łukasz Bibrzycki** [4],
**Michalina Borczyńska** [5], **Łukasz Bratek** [6], **Nikolay Budnev** [7], **Dariusz Burakowski** [8],
**David E. Alvarez-Castillo** [1,9], **Kevin Almeida Cheminant** [1], **Aleksander Ćwikła** [10],
**Punsiri Dam-o** [11], **Niraj Dhital** [12], **Alan R. Duffy** [13], **Piotr Głownia** [6],
**Krzysztof Gorzkiewicz** [1], **Dariusz Góra** [1], **Alok C. Gupta** [14], **Zuzana Hlávková** [15],
**Martin Homola** [15], **Joanna Jałocha** [6], **Robert Kamiński** [1], **Michał Karbowiak** [16],
**Marcin Kasztelan** [17], **Renata Kierepko** [1], **Marek Knap** [18], **Péter Kovács** [19], **Szymon Kuliński** [6],
**Bartosz Łozowski** [20], **Marek Magryś** [21], **Mikhail V. Medvedev** [22,23], **Justyna Mędrala** [24],
**Jerzy W. Mietelski** [1], **Justyna Miszczyk** [1], **Alona Mozgova** [25], **Antonio Napolitano** [26],
**Vahab Nazari** [1,9], **Y. Jack Ng** [27], **Michał Niedźwiecki** [10], **Cristina Oancea** [28,29], **Bogusław Ogan** [30],
**Gabriela Opiła** [24], **Krzysztof Oziomek** [21], **Maciej Pawlik** [21,22], **Marcin Piekarczyk** [4],
**Bożena Poncyljusz** [5], **Jerzy Pryga** [31], **Matías Rosas** [32], **Krzysztof Rzecki** [24],
**Jilberto Zamora-Saa** [33], **Katarzyna Smelcerz** [10], **Karel Smolek** [34], **Weronika Stanek** [24],
**Jarosław Stasielak** [1], **Sławomir Stuglik** [1], **Jolanta Sulma** [35], **Oleksandr Sushchov** [1],
**Manana Svanidze** [36], **Kyle M. Tam** [37], **Arman Tursunov** [38], **José M. Vaquero** [39],
**Tadeusz Wibig** [16] and **Krzysztof W. Woźniak** [1]

[1] Institute of Nuclear Physics Polish Academy of Sciences, 31-342 Kraków, Poland;
   david.alvarez@ifj.edu.pl (D.E.A.-C.); kevin.almeida-cheminant@ifj.edu.pl (K.A.C.);
   krzysztof.gorzkiewicz@ifj.edu.pl (K.G.); Dariusz.Gora@ifj.edu.pl (D.G.); Robert.Kaminski@ifj.edu.pl (R.K.);
   Renata.Kierepko@ifj.edu.pl (R.K.); jerzy.mietelski@ifj.edu.pl (J.W.M.); Justyna.Miszczyk@ifj.edu.pl (J.M.);
   vnazari@jinr.ru (V.N.); jaroslaw.stasielak@ifj.edu.pl (J.S.); slawomir.stuglik@ifj.edu.pl (S.S.);
   oleksandr.sushchov@ifj.edu.pl (O.S.); Krzysztof.Wozniak@ifj.edu.pl (K.W.W.)
[2] Department of Chemistry and Physics, Clayton State University, Morrow, GA 30260, USA;
   dmitriybeznosko@clayton.edu
[3] Astronomical Observatory, Jagiellonian University, 30-244 Kraków, Poland; gopal@oa.uj.edu.pl
[4] Institute of Computer Science, Pedagogical University of Krakow, 30-084 Kraków, Poland;
   lukasz.bibrzycki@up.krakow.pl (Ł.B.); marcin.piekarczyk@up.krakow.pl (M.P.)
[5] Faculty of Physics, University of Warsaw, 02-093 Warsaw, Poland; m.borczynska4@student.uw.edu.pl (M.B.);
   b.poncyljusz@student.uw.edu.pl (B.P.)
[6] Institute of Physics, Cracow University of Technology, PL-30084 Kraków, Poland;
   Lukasz.Bratek@pk.edu.pl (Ł.B.); piotr.glownia@student.pk.edu.pl (P.G.); joanna.jalocha-bratek@pk.edu.pl (J.J.);
   szymon.kulinski@student.pk.edu.pl (S.K.)
[7] Irkutsk State University, 664003 Irkutsk, Russia; nbudnev@api.isu.ru
[8] Astroparticle Physics Amateur, 30-322 Kraków, Poland; buriaszany@gmail.com
[9] Joint Institute for Nuclear Research, Joliot-Curie street 6, 141980 Dubna, Russia
[10] Department of Computer Science, Faculty of Computer Science and Telecommunications,
   Cracow University of Technology, 31-155 Kraków, Poland; aleksander.cwikla@student.pk.edu.pl (A.Ć.);
   nkg@pk.edu.pl (M.N.); katarzyna.smelcerz@pk.edu.pl (K.S.)
[11] School of Science, Walailak University, Thasala 80160, Thailand; dpunsiri@mail.wu.ac.th
[13] Department of Physics, St. Xavier's College, Maitighar, Kathmandu 44600, Nepal; ndhital@mtu.edu
[13] Centre for Astrophysics and Supercomputing, Swinburne University of Technology,
   Melbourne, VIC 3122, Australia; aduffy@swin.edu.au
[14] Aryabhatta Research Institue of Observational Sciences (ARIES), Manora Peak, Nainital 263001, India;
   alok@aries.res.in
[15] Faculty of Mathematics, Physics and Informatics, Comenius University in Bratislava, Mlynská Dolina,
   842 48 Bratislava, Slovakia; hlavkova14@uniba.sk (Z.H.); homola@fmph.uniba.sk (M.H.)
[16] Faculty of Physics and Applied Informatics, University of Łódź, 149/153 Pomorska, 90-236 Łódź, Poland;
   michal.karbowiak@fis.uni.lodz.pl (M.K.); t.wibig@gmail.com (T.W.)
[17] National Centre for Nuclear Research, Andrzeja Sołtana 7, 05-400 Otwock-Świerk, Poland; mk@zpk.u.lodz.pl

[18]    Astroparticle Physics Amateur, 58-170 Dobromierz, Poland; mpknap@wp.pl

[19]    Institute for Particle and Nuclear Physics, Wigner Research Centre for Physics, Konkoly-Thege Miklós út 29-33, 1121 Budapest, Hungary; kovacs.peter@wigner.hu

[20]    Faculty of Natural Sciences, University of Silesia in Katowice, Bankowa 9, 40-007 Katowice, Poland; bartosz.lozowski@us.edu.pl

[21]    ACC Cyfronet AGH-UST, 30-950 Kraków, Poland; M.Magrys@cyfronet.pl (M.M.); k.oziomek@cyfronet.pl (K.O.); m.pawlik@cyfronet.pl (M.P.)

[22]    Department of Physics and Astronomy, University of Kansas, Lawrence, KS 66045, USA; medvedev@ku.edu

[23]    Laboratory for Nuclear Science, Massachusetts Institute of Technology, Cambridge, MA 02139, USA

[24]    AGH University of Science and Technology, 30-059 Kraków, Poland; medrala@student.agh.edu.pl (J.M.); gopila@student.agh.edu.pl (G.O.); krz@agh.edu.pl (K.R.); wstanek@student.agh.edu.pl (W.S.)

[25]    Astronomical Observatory of Taras Shevchenko National University of Kyiv, 04053 Kyiv, Ukraine; alenamozgova@ukr.net

[26]    University of Napoli "Parthenope", 80143 Napoli, Italy; antonio.napolitano@uniparthenope.it

[27]    Department of Physics and Astronomy, University of North Carolina, Chapel Hill, NC 27599, USA; yjng@physics.unc.edu

[28]    ADVACAM, 12, 17000 Prague, Czech Republic; cristina.oancea@advacam.com

[29]    University of Bucharest, 077125 Bucharest, Romania

[30]    Astroparticle Physics Amateur, 86-170 Nowe, Poland; boguslaw.ogan@outlook.com

[31]    Jagiellonian University, 31-007 Kraków, Poland; jerzy.pryga@wp.pl

[32]    Institute of Secondary Education, Highschool No. 65, 12000 Montevideo, Uruguay; mrosas@docente.ceibal.edu.uy

[33]    Departamento de Ciencias Fisicas, Universidad Andres Bello, Piso 7, Sazie 2212, Santiago, Chile; jilberto.zamora@unab.cl

[34]    Institute of Experimental and Applied Physics, Czech Technical University in Prague, Husova 240/5, 110 00 Prague, Czech Republic; karel.smolek@utef.cvut.cz

[35]    Publiczna Szkoła Podstawowa im. Św. Jadwigi Królowej w Rzezawie, 32-765 Rzezawa, Poland; jola.sulma@wp.pl

[36]    E. Andronikashvili Institute of Physics under Tbilisi State University, 0177 Tbilisi, Georgia; manana.svanidze@tsu.ge

[37]    Waterloo Rocketry, University of Waterloo, Ontario, N2L 3G1, Canada; kmjtam@uwaterloo.ca

[38]    Research Centre for Theoretical Physics and Astrophysics, Institute of Physics, Silesian University in Opava, Bezručovo nám. 13, CZ-74601 Opava, Czech Republic; arman.tursunov@physics.slu.cz

[39]    Department of Physics, University of Extremadura, 06800 Mérida, Spain; jvaquero@unex.es

**\***    Correspondence: Piotr.Homola@ifj.edu.pl; Tel.: +48-12-662-8341

**Abstract:** The Cosmic-Ray Extremely Distributed Observatory (CREDO) is a newly formed, global collaboration dedicated to observing and studying cosmic rays (CR) and cosmic-ray ensembles (CRE): groups of at least two CR with a common primary interaction vertex or the same parent particle. The CREDO program embraces testing known CR and CRE scenarios, and preparing to observe unexpected physics, it is also suitable for multi-messenger and multi-mission applications. Perfectly matched to CREDO capabilities, CRE could be formed both within classical models (e.g., as products of photon–photon interactions), and exotic scenarios (e.g., as results of decay of Super-Heavy Dark Matter particles). Their fronts might be significantly extended in space and time, and they might include cosmic rays of energies spanning the whole cosmic-ray energy spectrum, with a footprint composed of at least two extensive air showers with correlated arrival directions and arrival times. As the CRE are predominantly expected to be spread over large areas and, due to the expected wide energy range of the contributing particles, such a CRE detection might only be feasible when using all available cosmic-ray infrastructure collectively, i.e., as a globally extended network of detectors. Thus, with this review article, the CREDO Collaboration invites the astroparticle physics community to actively join or to contribute to the research dedicated to CRE and, in particular, to pool together cosmic-ray data to support specific CRE detection strategies.

**Keywords:** cosmic rays; cosmic-ray ensembles; extensive air showers; large scale cosmic-ray correlations; CREDO Collaboration

---

## 1. Introduction

Although state-of-the-art cosmic-ray research to date has been focused on the detection and analysis of cosmic particles observed through individual detectors or arrays, the correlated observations of cosmic rays on the global scale remain insufficiently explored, yet no less promising. This collaborative perspective could provide a deeper insight into the physical processes within energy ranges rarely considered, including the highest energies known. Here we discuss a general approach to research dedicated to detecting and studying the astroparticle physics phenomena called Cosmic-Ray Ensembles (CRE) defined as groups of a minimum of two correlated, be it spatially or temporally, cosmic rays with a common primary interaction vertex or the same parent particle. Such particles, constituents of CRE, are messengers of the primary physical processes—probes of the physics that happened even billions of years ago, at energies even millions of times higher than energies to which we can accelerate particles using man-made infrastructure.

Armed with the particle physics background telling us that cosmic-ray particles are expected to interact with radiation and matter on their way through the Cosmos and give birth to CRE, we ask not whether CRE exist, but under which circumstances and with which conditions they can reach the Earth and be detected with the available or possible infrastructure. The signatures of CRE might be spread over very large areas, as illustrated in Figure 1, and this feature might make them all but impossible to detect by existing detector systems operating in isolation. On the other hand, if the active detectors operate as part of a global network, as proposed by The Cosmic-Ray Extremely Distributed Observatory (CREDO) [1–3], these CRE are naturally more likely to be observed. The particles (including photons) that constitute a CRE might have energies that essentially span the entire cosmic-ray energy spectrum, so in practice all cosmic-ray detectors could contribute in a common effort towards the hunt for CRE. The list of useful devices stretches from smartphones (e.g., DECO [4], CRAYFIS [5], CREDO Detector [6–8], Cosmic Ray App [9]), to pocket-size open-hardware scintillator detectors such as Cosmic Watch [10,11] or CosmicPi [12], and numerous larger educational detectors and arrays as well e.g., HiSPARC [13,14], QuarkNet [15], Showers of Knowledge [16], or CZELTA [17]. Similarly, the data recorded by professional instruments that receive, or will receive, cosmic rays as a signal or as a background would be of importance for CRE-oriented strategies. Here the, non-exhaustive, list includes the Pierre Auger Observatory [18], Telescope Array [19], JEM-EUSO [20,21], HAWC [22], MAGIC [23], H.E.S.S. [24], VERITAS [25,26], IceCube [27], Baikal-GVD [28], ANTARES Telescope [29], European Southern Observatory [30], other astronomical and underground/underwater observatories, and also accelerator experiments in off-beam mode. Therefore, it is not only desirable, but also feasible to put the CRE research into a routine implementation. So far, the experimental searches for cosmic-ray correlations have been realized on different scales with arrays of detectors located at schools and universities. However, all those efforts were dedicated to very specific scenarios, concerning mostly fragmentation of nuclei in background electromagnetic radiation fields [31], which limits the number of particles in the group (ensemble) to just a few. Some of these projects include e.g., CHICOS [32] in the U.S., ALTA [33] in Canada, GELATICA [34] in Georgia, EEE [35] in Italy, LAAS [36,37] in Japan, and the aforementioned CZELTA in Czech Republic. Time correlation of registered showers was studied at distances from 100 m to 7000 km, and in some cases evidence for unexpected coincidences have been found. However, these were without any convincing follow-up studies and data taking campaigns, which is hard without global coordination. Only very recently the idea of looking for large scale correlations in a general and global way took shape with the CREDO Collaboration, formalized in September 2019 [38]. CREDO is meant to be a multi-technique (different detector types) and doubly open (for both data upload and offering access) infrastructure enabling global research programs

concerning radiation (both cosmic and terrestrial), with several multi-messenger, multi-mission and transdisciplinary opportunities. With this review article we invite the community to both benefit from the openness of CREDO and to contribute to its program with the research dedicated to a general search for ensembles of cosmic rays, especially photons of different energies. Since the concept of CREDO assumes the openness for independent data streams, it is expected that the projects mentioned above, as well as the other cosmic-ray infrastructure including private, widely spread detectors such as smartphones, will have a direct interest in connecting to the global CREDO system, thus reinforcing its scientific attractiveness, and chances of individual research programs.

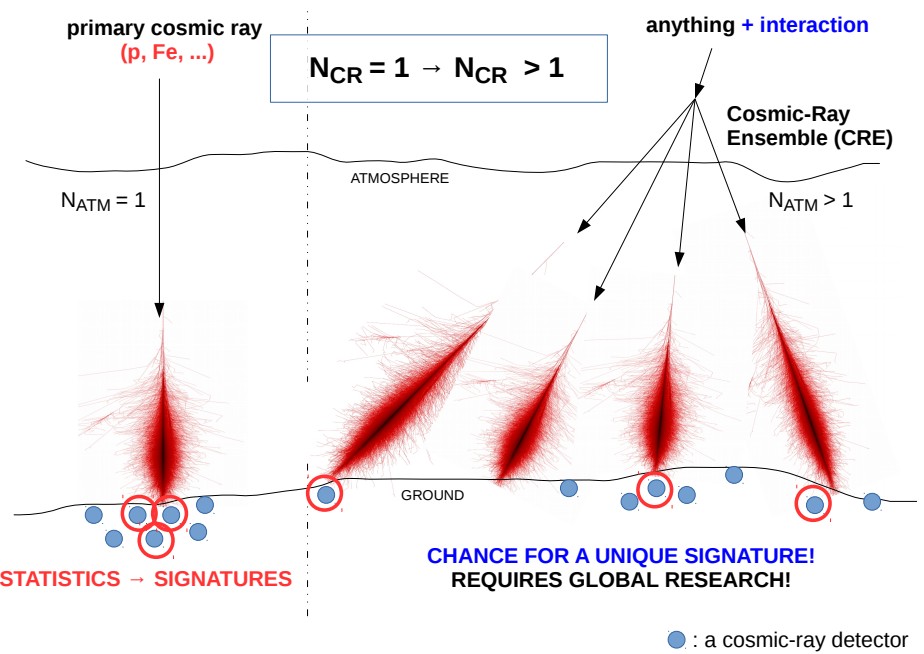

**Figure 1.** Cosmic-Ray Ensembles: a novelty in cosmic-ray research and in multi-messenger astroparticle physics.

As this article is meant to serve as a review of the current status and perspectives of CREDO and the research related to CRE, its structure follows the scientific and logical roadmap to observing and studying cosmic-ray ensembles. We begin with a general description of the field in Section 2 "Foundations of the CREDO methodology" and Section 3 "CREDO within the cosmic-ray landscape", and theoretical modeling of CRE sources in Section 4 "UHECR sources and cosmic-ray ensembles". Then we present and discuss example CRE scenarios with an emphasis on the simulations of propagation of the CRE components through the Cosmos and within the Earth's atmosphere (Section 5 "CRE simulations"), describe the status of the observational efforts (Sections 6 "CREDO detectors: cloud of clouds", Section 7 "Data management and analysis", and Section 8 "Building the scale: public engagement as a scientific need"), and conclude the article with the outlook, summary and conclusion (Sections 9 and 10).

## 2. Foundations of the CREDO Methodology

The CREDO experiment by its very idea of making discoveries and approaching truth in various research areas, touches something one can call the borderline of the current state of our knowledge. As explained below, there are reasons to anticipate that the experiment has the potential to falsify some of the adopted models. In the context of CREDO one can even ask questions about science itself and its methodology.

As an example, one can consider exotic QED processes which are potentially within the observational scope of CREDO. Specifically, pair creation or photon splitting in strong magnetic fields. These predictions of QED can now be tested in the context of cosmic rays physics under extreme astrophysical conditions met in pulsar magnetospheres (with typical magnetic field strengths of $10^{12}$ Gauss, or even $10^{14}$ Gauss in magnetars). In such conditions there are several observational signatures of the two processes [39,40]. Furthermore, the photon splitting phenomenon (in exotic scenarios) fits very well in the context of cosmic-ray ensembles (CRE) that are potentially within the observational scope of CREDO (as long as the opening angle of the secondary photons would not be too large). Therefore, the CREDO experiment opens up new opportunities to test the well-established QED theory as well as consider more exotic scenarios. The experiment is potentially capable of changing our view of the basis of science, specifically by providing a new type of means to falsify generally accepted theories in an energy range that has not been yet available for to date observatories. In this way one arrives at the strictly philosophical question to be addressed and that naturally arises in the context of CREDO—the issue of parsimony of the scientific method or Occam's razor.

It has been accepted that the scientific method of explaining and understanding facts should follow the principle of not multiplying entities without necessity. Explaining all new phenomena should be based on what one has already had, specifically based on the theories and models that successfully worked so far, and only if this attempt is found to have failed one may consider rejecting these theories and models. Occasionally, the parsimony principle, when understood in a fundamentalistic manner, would stand in the way of knowledge. This kind of epistemological attitude might lead to supplementing old models and theories with new "epicycles", rather than encouraging one to humbly admit that some of the assumptions being adopted so far need to be rejected as inconsistent with reality. A solution to this state of things is to presume that the parsimony principle, although an invaluable component of the scientific method, is not something the scientific method entirely stands on. Sometimes one must bypass the strict rule (which in practice is done by most of the scientific community). Accordingly, if the CREDO experiment (or some other experiment) comes up with an anomalous result, not falling within the framework of accepted models and theories, first the possibility of an error should be considered, be it in the measurement or interpretation side. However, one should also not be afraid of pushing the limits and going beyond the realm of modern knowledge by extending or redefining the adopted concepts as well as the language with which we describe reality.

The CREDO initiative inspires one to deliberate on falsifiability, another important element of the scientific method. It is often said that scientific statements should be falsifiable—refutable by contradicting evidence. This is not definitely true. For example, general statements about existence are most often unfalsifiable, nevertheless they are easy to be verified empirically, which makes the statements quite scientific. Even though science is not entirely determined by falsifiability, this attribute is very advantageous and often characterizes statements formulated by science, either as models or scientific theories.

Classical electrodynamics provides a good example of a theory that is falsifiable. For illustration imagine that CREDO (or another experiment of this kind) comes up with some evidence for spontaneous photon splitting (to be in touch with the experimental scope of CREDO pointed out earlier)—a phenomenon there is no place for in the linear Maxwell electrodynamics. There are effective terms that could be incorporated to the electromagnetic Lagrangian to mimic at the classical level such effects. To be more specific, some QED vacuum polarization effects such as photon–photon scattering or photon splitting in external fields can be effectively described by classical theory (the Euler–Heisenberg Lagrangian is used in this context); however, these non-linear phenomena are merely interaction effects arising upon quantization of linear Maxwell fields in accordance with methods of quantum field theory, therefore already explainable within the current paradigm. The real change in the electromagnetic theory required to incorporate new phenomena such as spontaneous photon splitting that could occur in free space and observed through cosmic-ray ensemble evolution, would be to replace the Maxwell Lagrangian by an alternative one resulting in non-linear equations before quantization, or maybe even

to alter the quantization method as such. Is that to mean that Maxwell's electrodynamics would just be falsified by an observation of a spontaneous photon splitting?

Stated clearly, it is not that simple—scientific theories are deeply founded, routinely withstanding repeated attempts at falsification. In the first place, one should try to explain a particular observational result within the current theory. Only if such attempts turn out absolutely unsuccessful and a growing number of anomalies are observed at the same time, one would rather conclude that the theory has been falsified—though the old theory will probably still survive as a good approximation (as this was the case with Newtonian and relativistic mechanics). Thus, falsifying a theory is not an easy task, at least not possible-based merely on a single observation or measurement. This being said, experiments such as CREDO appear even more valuable, since both for the verification or falsification to be feasible, one needs as many channels as possible through which the universe is looked at.

## 3. CREDO within the Cosmic-Ray Landscape

Although CRE detectable on or around the Earth can be initiated by particles spanning a wide range of energies, and since it is not a priori evident whether the chances of observing a CRE increase with the energy of the primary particle, one should stay open-minded about possible focus concerning the energy regimes of investigation. Here, for clarity, we chose to focus on the cosmic rays of the highest energies known, $E > 10^{18}$ eV, hereafter referred to as ultra-high-energy cosmic rays (UHECR), capable of initiating CRE composed of billions of component particles which can propagate unaffected large astrophysical distances, as in case of GeV-TeV photons (see e.g., [41]), of which an observable fraction may reach the human technosphere.

The surprisingly constant, power-law character of the energy spectrum of cosmic radiation observed by more than ten orders of magnitude with an almost constant exponent of about $(-3)$ could be, in principle, continued on without any obstacles. At least until the mid-1960s there was no reason to expect any definite end. Although the sources and mechanisms of acceleration of single elementary particles to the energy of a dozen joules both then and now are unknown (the search is still ongoing). Review of objects in the sky that would potentially come into question, their spatial sizes ($L$) and magnetic fields ($B$) suggests that their capabilities ($E_{max} < ZeBL$) do not reach far beyond the limit of $10^{20}$ eV. It should be remembered that any acceleration mechanism by its nature would have to be of a statistical character and therefore, when analyzing the average or typical parameters of primary cosmic-ray particles, it is difficult to exclude the occurrence of large and extremely large fluctuations [42].

The situation changed substantially after the discovery of the cosmic microwave background (CMB) radiation. If we assume that the primary particles of cosmic radiation are protons, as Greisen [43], Zatsepin and Kuzmin [44] noticed immediately, a sudden end of the spectrum must be caused by collisions of them with CMB photons and resonant production of the $\Delta^+$ particle. $\Delta^+$ decays instantly again into a nucleon, and its energy is on average about 20% less than before the collision. This process occurs as long as the nucleons have enough energy, which happens at about 5–6 $\times 10^{19}$ eV. However, particles of cosmic radiation do not have to be protons. Observations suggest (e.g., [45]) that heavier nuclei with a higher charge ($Ze$), thus easier to be accelerated, may entirely dominate the highest energy cosmic radiation flux. For nuclei with energies of about $10^{20}$ eV, when energies are calculated separately for each nucleon, the GZK mechanism does not work, but this does not mean that the Universe for them remains transparent. In the center of mass system, when colliding with photons of intergalactic radiation (infrared mostly) they excite to a giant dipole resonance, and then fragment, most preferably emitting few neutrons. Although fragments retain the same energy per nucleon, as a whole they have correspondingly reduced their total energy.

Searching for the end of the cosmic radiation spectrum, according to what has been just said, could lead to solving the problem of mass composition of cosmic radiation of the highest energy, and thus bring us closer to identifying its sources (and acceleration mechanisms). However, it unfortunately happens that both the GZK cut-off and the photodisintegration of the nuclei start

to work effectively in a similar range of energy, just where today's observations of cosmic radiation end because of the scarce cosmic-ray flux. In addition to the statistics, another circumstance is that two giant experiments in operation today which statistically dominate the measurements of giant showers in the highest energy range: the Pierre Auger Observatory in Argentina and the Telescope Array in Utah, US, while claiming a general agreement of the energy spectra within experimental uncertainties up to 10 EeV, they admit the need for non-linearity to bring the spectra in agreement at the higher energies, and within the range of common declinations. However, the sources of this non-linearity have not been identified, yet [46]. It turns out, then, that the current status of the UHECR observations does not allow definite conclusions about the exact location and nature of the observed cut-off of the energy spectrum, implying the uncertainty about the cosmic-ray composition at the highest energies known.

Either way, the results of the leading UHECR observatories and their conclusions from widely quoted publications ([47–50]) indicate that the spectrum of cosmic radiation is significantly suppressed at the highest energies, although it is not known at which energy, and whether this energy depends e.g., on the kind of the sources. Such a picture is widely accepted and despite minor scratches, small inconsistencies and doubts it seems that we understand it. There should not be many particles above the cut-off and in fact this is the case. However, among the aforementioned minor doubts, there are still recorded in the last century in several (or actually almost all) large and significant giant air shower experiments, the cases initiated by particles with estimated energies exceeding $10^{20}$ eV.

The first, historical, Volcano Ranch event was recorded by Linsley in 1962 with the energy of $10^{20}$ eV, [51]. In the 1980s the Haverah Park experiment reported a significant increase in the number of showers with energies exceeding $\sim 10^{20}$ eV [52–54] . In the Yakutsk array, the record energy was rated at $1.5 \times 10^{20}$ eV [55]. The Japanese AGASA experiment published the event from 3 December 1993 which had an energy of $2 \times 10^{20}$ eV [56]. However, a world record was set in the USA in the Fly's Eye experiment: $3.2 \times 10^{20}$ eV [57]. These cases have not been discussed recently in the literature, but it seems that they require some explanation. The first, more straightforward explanation is that the experimenters might have been misled about the energy reconstruction of their record event by the imperfectness of the tools available at that time, with particular emphasis on numerical tools. Thankfully, the Monte Carlo methods developed in the 21st century are certainly more precise and they allow making more adequate corrections today than decades ago, in particular in experimental procedures such as localization of shower axis, or estimation of shower energy, whether by collecting fluorescent light or determining the charged particle density distribution on the ground. In addition, this explanation could be enough, provided one does not discuss in detail the statistics of the "above $10^{20}$ eV" cosmic-ray detections of 20th century experiments in contemporary analyses. However, as already mentioned, the two great experiments, the Pierre Auger Observatory and the Telescope Array, also reported around 20 UHECR cases of energies above $10^{20}$ eV observed already with the newest tools and methods (see e.g., Ref. [58] where 15 events with energies above $10^{20}$ eV are mentioned collectively). To list just a few such events in detail we mention the Pierre Auger Observatory measurements which contain an event with energy $1.4 \times 10^{20}$ eV [59] – or $1.3 \times 10^{20}$ eV [60], and the Telescope Array announcement concerning an event of similar energy $1.39 \times 10^{20}$ eV [48]. It is of course clear that if the cosmic-ray energy spectrum breaks around $5$–$6 \times 10^{19}$ eV, this is most likely a (gradual) suppression rather than an abrupt cut-off, so there must be a few events exceeding $10^{20}$ eV. However, the current statistics of these events does not allow telling whether they are compatible with the spectrum cut-off or not.

It is a great achievement that today we can tell that the UHECR spectrum has been quite well measured—see for example two recent Pierre Auger Observatory papers [58,61], and that a lot of effort and resources are being currently invested into explaining the nature of the observed spectrum suppression (GZK-like versus acceleration limit). Nevertheless the current results are still inconclusive, and it is, therefore, advisable to continue research in the highest energy regime of cosmic rays, and to try alternative methods whenever possible. In other words, the status of the dispute in the UHECR

area encourages a closer look at the field and being ready for a major revision or breakthrough in the understanding of physics at the highest energies known. The CREDO initiative with its objectives dedicated to going beyond studying individual cosmic rays and taking under investigation also UHECR products—cosmic-ray ensembles, may provide a precious complementary approach to UHECR studies.

## 4. UHECR Sources and Cosmic-Ray Ensembles

The Standard Model (SM) of particle physics suggests that if cosmic rays (CR) are mostly protons, there must exist a cut-off in the energy spectrum of CR arriving from distances beyond ∼100 Mpc. This limit is a direct consequence of the GZK effect, originally presented in [43,44] and later improved in [62]. In direct contradiction to this, UHECR with energies greater than the GZK cut-off have been reported by experimentalists, from directions where no sources exist sufficiently nearby (The current experimental data suggests an extragalactic origin for UHECR with energies above the GZK cut-off [63–65].) Therefore, there are clear indications that our understanding of UHECR sources, nature, and/or propagation is incomplete or even flawed. On the other hand, the CR background has been studied by means of numerical codes [66,67], providing solid arguments that support the idea that CR with energies over the GZK cut-off *cannot* be photons.

In general, one can distinguish two qualitatively different approaches in unveiling the physics of UHECRs: theoretical models assuming interactions of exotic super-heavy matter (including extra dimensions, Lorentz invariance violation, cosmic strings, existence of new particles etc. [68–85]) and acceleration scenarios describing processes, in which the particles are accelerated by a particular astrophysical object (shocks in relativistic plasma jets, unipolar induction mechanisms, second-order Fermi acceleration, etc.). Acceleration scenarios rely on the existence of powerful astrophysical sources with available energy that is sufficient for the energy transfer from these objects to cosmic-ray particles.

In the age of multi-wavelength and multi-messenger astronomy, transient astronomical objects are of the great interest for UHECRs emission. There are several classes of astronomical objects which can be prime targets for UHECR observations e.g., gamma-ray bursts (GRBs); supernovae (SN); fast radio bursts (FRBs); various classes of active galactic nuclei (AGN) such as Seyfert galaxies, radio galaxies and blazars; and possible neutrino emitting blazars. In the recent past there is evidence that these objects emit or can emit UHECRs. A 5-millisecond bright fast radio burst of extragalactic origin was detected [86]. It is claimed that the radio galaxies have emitted UHECRs [87]. There is evidence of hadronic $\gamma-$ray emission from supernova remnants [88]. Blazars are one of the most prominent sources of UHERCs emission. There are several papers which estimated/predicted neutrino and UHERCs emission from blazars (e.g., [89,90], and references therein). About 2 decades ago, it was predicted that there were some very bright high energy peaked blazars which would be neutrino loud [91]. And recently, the IceCube Collaboration found the evidence of neutrino emission from the blazar TXS 0506+056 [92] which opened a new window to search for such another blazars too.

### 4.1. Supermassive Black Holes as UHECR Sources

Among the most powerful astrophysical sources, one can highlight supermassive black holes (SMBHs) located in the centers of most galaxies due to their compactness and enormous energy available for the extraction. Below we shall briefly show the capability of SMBHs to power the UHECRs in a given model. It appears that up to 29% of the total energy ($M_{\mathrm{BH}}c^2$) of the black hole is rotational energy and available for extraction [93]. Presently, all tests of general relativity indicate that astrophysical black holes can be fully characterized by their masses and spins, while other properties of black holes are hidden inside the event horizon and unavailable to the external observer. For a typical SMBH of $10^9$ solar masses the extractable energy is of the order of $10^{74}$eV, turning SMBHs into the Universe's largest and the most compact energy reservoirs. Therefore, it is important to search for the processes, which tap these enormous energy sources in the most efficient way.

Attempts to tap the energy of black holes started in 1969 by Roger Penrose [94], followed by many authors (see, e.g., references [95–98] and references therein), who used the existence of negative energy states of particles with respect to a stationary observer at infinity. If a particle decays into two fragments near a black hole with one of the fragments attaining negative energy, the other fragment (respecting energy conservation) may escape from the black hole with greater energy than those of the primary particle. The remarkable property of the rotating black hole is the existence of a special region outside the event horizon called the ergosphere, where the energy of a particle relative to infinity can be negative. However, the negative energy states may also appear due to purely electromagnetic interactions between the SMBH and surrounding matter without the need for ergosphere [99,100].

Black holes, as any astrophysical object, are immersed into magnetic fields. Rotation of the black hole in an external magnetic field leads to the twisting of magnetic field lines due to the frame-dragging effect. Similar to a classical Faraday unipolar inductor, a black hole rotating in a magnetic field attains non-zero induced electric charge [101]. This charge has been shown to be non-negligible for any astrophysical black hole and is potentially observationally measurable [102]. Since the induced charge of the black hole is coupled and proportional to the black hole spin, its discharge is equivalent to the slowing down of the black hole and decreasing the black hole's rotational energy. To demonstrate the process of black hole energy extraction and resulting acceleration of particles to ultra-high energy, one can consider the ionization or decay of a neutral particle into charged fragments in the vicinity of rotating SMBH immersed into an external magnetic field. If the black hole possesses induced electric charge, the energy of charged fragments after decay of the neutral particle obtains a strong Coulombic contribution in addition to the gravitational negative energy in the ergosphere. Since the induced charge of SMBHs is more likely positive [102], these are the protons, which would escape from the black hole with tremendous energy [100]. This is the ultra-efficient regime of the so-called magnetic Penrose process that can serve as a possible mechanism for acceleration of UHECRs when applied to SMBH candidates. The energy of an escaping UHE particle depends on its charge to mass ratio, strength of external magnetic field and the mass of the black hole. For protons accelerated in this mechanism, the maximum energy is predicted to be

$$E_{\mathrm{p}} = 1.3 \times 10^{21} \mathrm{eV} \, \frac{Z \, m_p}{m} \, \frac{B}{10^4 \mathrm{G}} \, \frac{M}{10^{10} M_\odot}. \tag{1}$$

In Figure 2 we demonstrate the acceleration of UHE protons resulting from the hydrogen ionization in the SMBH vicinity. Similar results are obtained for the neutron beta decay ($n \rightarrow p^+ + e^- + \bar{\nu}_{\mathrm{e}}$). Here we provide verifiable constraints on the SMBH mass and magnetic field as UHECR source. As an example we indicate the few famous nearby SMBH candidates, which can produce UHE protons of certain energies. It is interesting to note that the black hole located at the center of our Galaxy can accelerate particles up to the energies corresponding to the knee of the cosmic-ray spectrum. On the right side of Figure 2 we demonstrate the results of numerical simulation of the magnetic Penrose process (for details, see [99,100,103]). Remarkably, the mechanism operates in viable physical conditions for typical SMBHs, without the need for a large acceleration zone or exotic assumptions for black holes.

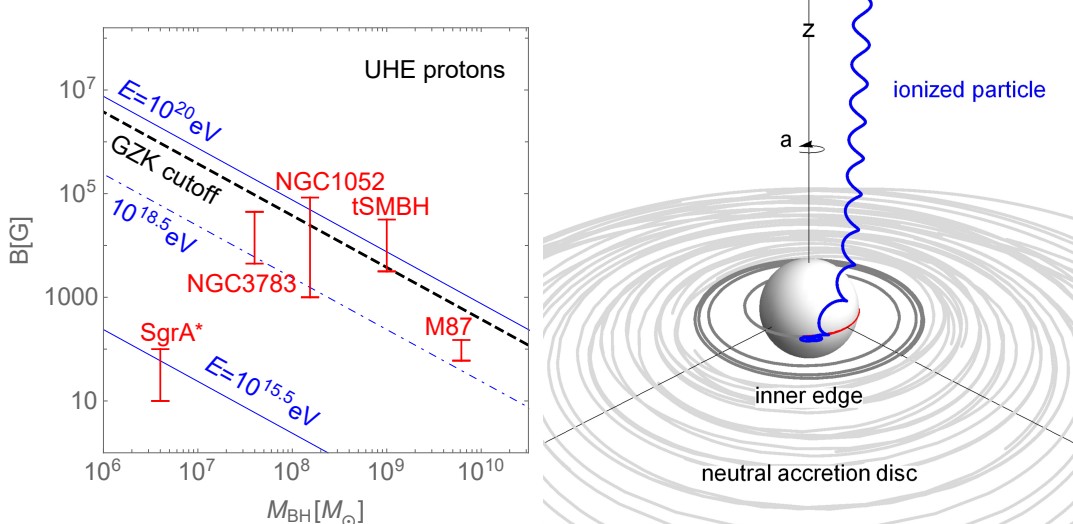

**Figure 2.** *Left:* constraints on the SMBH mass and magnetic field for UHE protons and chosen nearby sources (at the distance < 20 Mpc from the Solar system) fitting UHECRs of certain energy. Data is taken from [104–109], tSMBH source corresponds to a typical SMBH of mass $10^9 M_\odot$ and magnetic field $10^{4\pm0.5}$ G. *Right:* numerical simulation of a decay (ionization) of a freely falling neutral particle (grey thick) into two charged fragments in the vicinity of a rotating black hole immersed in an external magnetic field. Positively charged fragment (blue) is accelerated by the black hole and escapes to infinity along the symmetry axis. Negatively charged fragment (red) ultimately falls into the black hole, extracting its rotational energy (see, details in [100].)

### 4.2. Axion-Like Particles as UHECRs

A different approach to explaining the observation of UHECR avoiding the GZK cutoff is to propose particles beyond the SM. In such a case, the UHECR must be composed of new particles that fulfill the following conditions: (i) have a long mean-life in order to reach the Earth from a cosmological distance without decay; (ii) have a strongly suppressed coupling with photons in order to avoid energy loss through interaction with the CMB and galactic magnetic fields; (iii) be produced profusely at the source; and (iv) interact strongly enough in the proximity of Earth.

Among the new particles that may fulfill the above conditions it is worth mentioning axions which were originally proposed by Peccei and Quinn [110,111] in order to solve the strong CP problem and later proposed as candidates to avoid the GZK cut-off (see Refs. [112,113]). However, a recent study has shown that axion production, in addition to conversion into photons during their propagation through galactic magnetic fields, barely satisfies the present exclusion limits [114].

In recent years, the axion model has seen a resurgence, this time with the inclusion of new particles called Axion-Like-Particles (ALPs) which have similar features to axions. The Lagrangian for the most general ALPs is given by

$$\mathscr{L}_{\text{ALP}} = \frac{1}{2}\partial_\mu A\,\partial^\mu A - \frac{1}{2}m_A^2 A^2 - \frac{g_{\text{ALP}}}{4}aF_{\mu\nu}\tilde{F}^{\mu\nu}, \tag{2}$$

where $A$ ($m_A$) is the ALPs field (mass), the parameter $g_{\text{ALP}}$ is a model-dependent coupling constant which determines the interaction between ALPs and photons, $F_{\mu\nu}$ and $\tilde{F}_{\mu\nu}$ are the electromagnetic field strength and its dual, respectively.

The above-mentioned Lagrangian of Equation (2) includes a decay channel for ALPs into photons ($A \to \gamma\gamma$), which plays the role of a smoking-gun in experimental searches. The ALP decay width into two photons is presented in Refs. [115,116] as

$$\Gamma_A\,(A \to \gamma\gamma) = \frac{g_{\text{ALP}}^2 m_A^3}{64\pi}. \tag{3}$$

In addition to the ALPs decays, there is another peculiarity hidden in the above Lagrangian. ALPs can be converted into photons and photons into ALPs, in a similar way to neutrino flavor oscillation [117]. This peculiarity is known as the Primakoff effect [118]; and only occurs when ALPs are propagating in a strong external magnetic field. Each ALP-photon oscillation changes the polarization of the photon, providing an additional mechanism for an ALP's detection [119]. Within an astronomical context, the ALP-photon oscillation could lead to an apparent dimming of distant sources affecting the luminosity-redshift relation of Type Ia supernovae, the dispersion of quasar spectra and the spectrum of the CMB [112].

On the other hand, an ALP's propagation is not significantly affected by the CMB (due to very suppressed interaction between ALPs and photons) allowing an ALP to travel, without decaying, distances as large as the observed Universe's radius ($R_U$). The decay length of an ALP ($\lambda_A$) is given by

$$\lambda_A = \frac{E_A}{\Gamma_A m_A}, \tag{4}$$

where $E_A$ is the ALP energy. For ALPs to reach the Earth from a distance beyond the GZK limit, they must satisfy that $\lambda_A \lesssim R_U$. This scenario was studied in Ref. [114], leading to the following restriction on the ALPs decay width

$$\Gamma_A \lesssim \frac{E_A}{R_U m_A}. \tag{5}$$

and consequently, to a restriction on the ALPs coupling

$$g_{\text{ALP}} \lesssim \left( \frac{64\pi E_a}{R_U m_a^4} \right)^{1/2}. \tag{6}$$

The constraints on the ALP's parameter space, using the latest current experimental data, can be found in Refs. [113,114,120].

### 4.3. Dark Matter as a Source of UHECR

Two long-standing problems of contemporary astrophysics can be formulated as: (i) What is the nature of Dark Matter? and (ii) What is the nature of cosmic rays with energies above $10^{20}$ eV? Are these two questions related? The answer is not known. Yet, it is possible that these two great mysteries of modern astrophysics and cosmology can be explained within a single scenario. The concept of "*Occam's razor*" would say that a hypothesis which uses the smallest number of possible solutions to a collection of seemingly independent problems is the most preferred. Thus, a hypothesis that Super-Heavy Dark Matter (SHDM) and UHECR are related is worthy of investigation.

It is hypothesized that DM consists of SHDM with masses above $10^{23}$ eV, which could be produced in the early universe, e.g., before or during the inflation phase. Such SHDM may decay or be destroyed via annihilation [121] at the present epoch, leading to the copious production of less massive secondaries, or even predominantly photons. These secondaries can naturally have their energies at around $10^{20}$ eV or above. Note, it is hard to reconcile such UHECR energies with standard electromagnetic acceleration in any potential astrophysical source.

The SHDM hypothesis for the origin of UHECR predicts that the observed UHECR flux should mostly be dominated by energetic photons [122]. However, the highest energy events observed by two leading cosmic-ray observatories, namely the Telescope Array and Pierre Auger Observatory, do not consider them as photon candidates based upon their state-of-the-art air shower reconstruction techniques. Moreover, the absence of photon candidates above $10^{18}$ eV, favored by SHDM scenario, puts very stringent limits on it [123–125].

However, any conclusion about the absence of the photon candidates among UHECR and the subsequent conclusions about the SHDM hypothesis must be treated with caution. There are two main concerns as follows. First, the present analysis excludes from consideration any possible mechanism

that can mimic a hadronic air shower with a shower initiated by a photon. Such mechanisms could include a very efficient splitting of the energy of the primary into secondary photons/particles and underestimation of photonuclear interaction cross-sections. Second, the present analysis does not take into account any propagation effects of UHE photons. For example, it is not currently considered that very-high or ultra-high-energy photons might experience efficient screening or cascading on their way to the Earth, e.g., through interactions under special cases of Lorentz invariance violation [82]. In such a case, the products of this screening or cascading can spread over large areas and thus be far beyond the reach of the currently operating CR detectors. Such a non-observation may in turn be interpreted as an absence of UHE photons.

If the first possibility occurs, this would mean that UHE photons might have already been detected and they might be present in the data but not properly identified. If the second possibility—namely that the real properties of cosmic rays and the relevant propagation mechanisms are not well understood—takes place, then one has little or no chance to detect most of the very-high-energy photons that travel towards us. Both possibilities question the conclusion about limitations imposed on the SHDM scenarios by the presently accepted upper limits to photon fluxes. Furthermore, such a conclusion can be accepted only if both concerns above are addressed. A detailed study of these issues in the coming years is critical.

An inherent aspect of the SHDM scenario is the very existence of such DM particles. It is not an overstatement that science has currently no clue of what DM particles are, nor what their properties would be.

For a long time, the Weakly Interacting Massive Particles (WIMPs) were the most favored candidate. There was a reason for that—the "WIMP Miracle". The number density of the particles that freeze out in the early universe in a certain epoch is set by the balance of the two rates: the production-annihilation rate $n_X \langle \sigma v \rangle$ and the Universe's expansion rate $H$ (the Hubble constant at that epoch). Please note that $H$ characterizes the temperature of plasma in the universe $T$, so the number density of particles which are in thermal equilibrium can drop exponentially fast with lowering temperature if particles are heavy, $m_X c^2 \gg k_B T$. On the other hand, the cross-section depends on the coupling constant (i.e., the type of interaction) and the particle's mass. Next, the mass density of DM in the universe depends on both the density and mass, $\Omega_X \propto n_X m_X$, where $\Omega_X$ is the ratio (at present) of the particle-"X" mass density to the critical density in the universe. Furthermore, the coupling constant describing weak interactions—the Fermi constant, $G_F \approx 1.1 \times 10^5$ GeV$^{-2}$ naturally introduces the new mass scale of about 100 GeV. Finally, if the cross-section is the weak cross-section and the particle's mass is about $m_X \sim 100$ GeV, then the abundance of "X" is $\Omega_X \propto \langle \sigma v \rangle^{-1} \sim m_X^2 / g_X^4$ and $g_X \sim 0.6$, so $\Omega_X \sim 0.1$. Thus, "X" is Dark Matter. This is the "WIMP miracle": particle physics independently predicts particles with the right mass density to be Dark Matter. In this scenario, $\langle \sigma v \rangle \simeq 3 \times 10^{-26}$ cm$^3$s$^{-1}$; since $v \leq c$, the gross-section should be $\sigma \geq 10^{-36}$ cm$^2$. Numerous ongoing direct-detection DM experiments have reached sensitivity levels that correspond to $\sigma \sim 10^{-44} - 10^{-45}$ cm$^2$ in the mass range of interest—from tens of GeV to a few TeV, without a statistically significant and reproducible detection. Thus, the WIMP miracle is at odds with experiment.

The dismissal of the most favorable WIMP miracle in Dark Matter theory has ignited great interest in alternative scenarios. At present, all bets regarding the non-standard DM scenarios are on the table. There is no deficit of the putative candidates, including the super-heavy dark sector candidates. Here are some examples, as follows. (1) Magnetic monopoles [126,127] are topological defects that naturally appear in Grand Unified Theories (GUT) and carry a magnetic charge. The natural mass scale for them is the GUT scale, $\sim 10^{25}$ eV. However, the actual mass of a monopole is not constrained and all mass scales above the one that can be probed in an experiment are considered. As they are topological defects, monopoles are copiously produced in a GUT phase transition, creating a severe over-closing problem $\Omega_X \gg 1$. This problem is remedied by inflation, which reduces the monopole abundance in the amount enough to not contradict observational and

recent cosmological constraints [128]. (2) Wimpzillas [129] are super-heavy DM particles, which mass scale is many orders of magnitude above the conventional WIMP scale, possibly as large as the GUT scale. The WIMP mass cannot exceed hundreds of TeV. Otherwise, heavier WIMPs would over-close the universe, $\Omega_X > 1$. Therefore, Wimpzillas, like monopoles, are not 'thermal relics'. They emerge out of thermal equilibrium right after inflation and their density is not determined by the balance of the production-annihilation rate and the universe expansion rate. (3) Planckian-scale particles [130–132] is a whole class of candidates that appears in a minimal scenario of DM, where only gravitational interactions with the standard model are assumed. There is only one parameter in the scenario—the particle mass, which ranges from TeV up to the GUT scale. These particles are assumed to be produced by gravitational scattering in the thermal plasma of the Standard Model sector after inflation. For example, the Kaluza–Klein excitation of the graviton in the string theory is one of the realizations of this scenario. The above discussion does not present a complete list of SHDM candidates, of course, but just a few popular examples. There are more theoretically predicted candidates.

The search for signatures of propagation and/or decay of particle primaries beyond the Standard Model is to be thorough and taken seriously. In this regard, the CREDO observatory can serve as an indirect search for a wide class of non-standard Dark Matter candidates. Specifically, super-heavy DM particles whose masses are above $10^{23}$ eV. Due to this enormous mass, such particles (if they exist) can be produced in the early Universe, e.g., around the inflation phase. Their decay or annihilation at the present epoch can lead to observable products in the UHE range [74]. The interest in such a SHDM scenario is supported by certain difficulties the electromagnetic theory faces in explaining UHECR acceleration, as discussed below in Section 4.4.

### 4.4. Constraint on Electromagnetic Acceleration of UHECR

Electrically charged UHECR traversing a region of size $R$ filled with magnetic field $B$ assumed to be uniform, loses its energy due to synchrotron radiative energy loss [133] according to

$$\frac{dE}{dx} = F_{rad} = -\frac{2}{3}\left(\frac{Ze}{Am_pc^2}\right)^4 B^2 E^2, \tag{7}$$

where $A$ and $Z$ are the atomic mass and charge of an accelerated nucleus, $m_p$ is the proton mass. The solution to this equation is trivial:

$$E = \left(E_0^{-1} + E_{cr}^{-1}\right)^{-1}. \tag{8}$$

Thus, for an arbitrarily large initial energy, $E_0 \to \infty$, the particle energy cannot exceed the 'critical energy' threshold:

$$E_{cr} = \frac{3}{2}\left(\frac{Am_pc^2}{Ze}\right)^4 \left(B^2 R\right)^{-1} \sim 3 \times 10^{16} \frac{(A/Z)^4}{B_G^2 R_{kpc}} \text{ eV}, \tag{9}$$

where $B_G$ is in Gauss and $R_{kpc}$ is in kiloparsec.

Furthermore, even if one is to continuously accelerate the particle along its curved path via an inductive electric force $F_{EM} = Ze\,E_{ind} \simeq Ze(v/c)B \sim Ze\,B$, one has the following energy evolution:

$$\frac{dE}{dx} = F_{EM} + F_{rad} = ZeB - \frac{2}{3}\left(\frac{Ze}{Am_pc^2}\right)^4 B^2 E^2. \tag{10}$$

It has a beautiful solution (assuming the initial energy is small compared to the final energy):

$$E = \sqrt{E_{acc}E_{cr}}\tanh\sqrt{E_{acc}/E_{cr}}, \tag{11}$$

where

$$E_{acc} = Ze\,B\,R \sim 9 \times 10^{23} Z\,B_G\,R_{kpc} \tag{12}$$

is the maximum energy the accelerated particle can reach without losses. This solution has two obvious asymptotic scalings. If $E_{acc} \ll E_{cr}$, i.e., losses are small, one recovers the Hillas constraint [134] $E \simeq E_{acc}$, and in the opposite limit one has

$$E_{\max} \simeq \sqrt{E_{acc} E_{cr}} \sim 10^{20} A^2 Z^{-3/2} B^{-1/2} \text{ eV.} \tag{13}$$

These results are summarized in Figure 3. All the curves, except for two dotted ones, correspond to protons ($A = Z = 1$), and the dotted are for iron nuclei ($A = 56, Z = 26$). The details are discussed in the figure caption. Here we make a few important conclusions. First, there is an absolute maximum energy of UHECR accelerated electromagnetically and subject to radiative losses (e.g., in dipolar magnetospheres, galactic and extragalactic shocks and such), which is given by the balance of the acceleration and losses. This puts the upper limit on the magnetic field strength, given by the dot-dashed horizontal line and by Equation (13):

$$B \lesssim E_{20}^{-2} A^4 Z^{-3} \text{ Gauss,} \tag{14}$$

where $E_{20} = E/(10^{20} \text{ eV})$. Second, there is a corresponding limit on the size of an accelerator. This is the size that corresponds to the "tip of the wedges", which can be obtained, for instance, from Equations (12) and (13), namely

$$R \gtrsim 6 \times 10^{-2} E_{20}^3 A^{-4} Z^2 \text{ pc.} \tag{15}$$

This constraint effectively rules out compact accelerators, such as neutron stars, white dwarfs and such. Large objects, such as galactic halos, radio lobes, and galaxy clusters become favorable. Third, electromagnetic acceleration of UHECR beyond $\sim 3 \times 10^{22}$ eV is hard because the size of an accelerator becomes comparable to the GZK distance. Furthermore, the acceleration beyond $\sim 3 \times 10^{23}$ eV would require an accelerator of the size of the observable universe, which is questionable. Fourth, our analysis above does not take into account that the source may be moving relativistically with a large Lorentz factor, as in a gamma-ray burst outflow, for example. Accounting for this, relaxes the size and field strength constraints but not very dramatically, as far as UHECR are concerned. Fifth, our analysis excludes special arrangements such as linear accelerators. If one can arrange particle acceleration along a straight path, e.g., strictly along a static straight magnetic field line, then the particle would experience no synchrotron energy loss, regardless of the field strength (still well below the QED, 'Schwinger field' strength). In this case, our analysis is inapplicable.

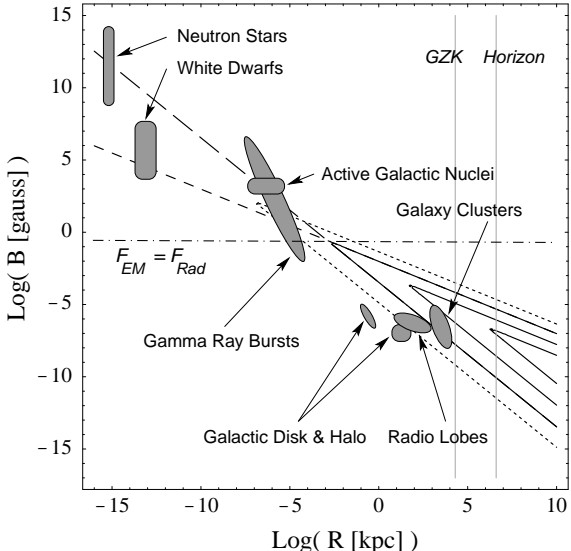

**Figure 3.** The well-known Hillas diagram, showing the magnetic field (*B*) versus object size (*R*) for UHECR acceleration [133]. The long-dashed line is the Hillas criterion, given by Equation (12), evaluated for a proton of energy $3 \times 10^{20}$ eV. The short-dashed and dot-dashed lines depict the radiative cooling constraints represented by Equations (9) and (13), and evaluated for a proton of $3 \times 10^{20}$ eV. In the region above the horizontal dot-dashed line, the radiative friction force exceeds the electromagnetic force. The solid lines show the boundaries of the allowed parameter regions. They are evaluated for protons of energies $3 \times 10^{20}$ eV, $10^{22}$ eV, and $3 \times 10^{23}$ eV, from the outermost to the innermost "wedge", respectively. The dotted lines are the same boundaries for an iron nucleus with energy $3 \times 10^{20}$ eV. Only those astrophysical objects which are located inside the "wedges" can, in theory, accelerate charged particles to such ultra-high energies. Finally, the grey vertical lines mark two characteristic spatial scales, namely (i) the GZK attenuation distance (about 20 Mpc) and (ii) the Hubble horizon scale (about 4 Gpc). For GRB sources, we took into account that the Lorentz boost changes with radius.

### 4.5. SQM Objects

Recently, strange quark matter (SQM) objects (either in the form of stars or planets) have been shown to efficiently convert mechanical energy into hadronic energy when they oscillate [135]. This is possible thanks to the property that the mass density at the edge of SQM objects must be precisely equal to the critical density of $4.7 \times 10^{14}$ g/cm$^3$, below which SQM is unstable with respect to decay into photons, hadrons, and leptons. Either compression or expansion of the SQM object, such as oscillation-induced deformations, releases energy. Such oscillations could be induced by tidal interactions with other bodies in stellar or planetary systems. Over a short time, of order 1 ms in a few oscillations the excitation energy would be converted into electromagnetic energy. With higher amplitude of the oscillations, the excitation energy would be released more rapidly and result even in fragmentation or dissolution of SQM objects. In the context of CREDO, it would be interesting to observe periodic signatures of such events.

In close encounters between an SQM object and another compact object, the energy transferred to the induced oscillations can reach $10^{53}$ erg for the closest approach distance (of 3 times the star radius). By coupling of oscillation modes, the monopole (that is, spherically symmetric radial) oscillations would also be excited. For violent encounters, the induced oscillations result in excitation energy in the surface layer region, be that an SQM star of pulsar mass or a planetary-mass SQM object. For fractional amplitudes $\chi = 10^{-6}$ of the radial oscillations, which are quite conservative, the corresponding deposited energy to be eventually radiated away is estimated to be $6.6 \times 10^{36}$ erg for a star of mass $1.4\, M_\odot$.

For more violent encounters (such as inspiraling of a tightly bound binary system), one can expect amplitudes as large as $\chi = 10^{-3}$ and the corresponding energy orders of magnitudes higher. A tightly bound binary system could induce periodic distortions of a SQM companion and periodic bursts of intense radiation (associated with gravitational radiation).

The energy loss due to the discussed excitation process should be accounted for in describing the evolution of a binary system with an SQM object, as this presumably would significantly change the predictions obtained with unexcited SQM objects. This is particularly important in the context of binary gravitational wave sources—more reliable template calculations should account for such excitations of SQM objects.

To understand the possible radiation scenarios, we present the following estimations taken from [135]. For small SQM objects called planets, the excited energy is deposited in the whole volume of the object. The energy then scales quadratically with the fractional amplitude of radial oscillations $\chi$ – for planets, the energy is $7.4 \times 10^{48} \cdot \frac{M}{M_\oplus} \chi^2$ erg. Accordingly, for an Earth mass SQM planet and $\chi = 10^{-5}$ one obtains $7.4 \times 10^{38}$ erg.

However, as numerical models demonstrate, the energy deposition zone shrinks towards the surface with increasing SQM object mass, and the energy law gradually changes. In the limit, for masses of order a solar mass, the deposition zone shrinks to a thin surface layer and the total excited energy can again be estimated in an analytical way; scaling cubically with the fractional amplitude of oscillations $\chi$ (the resulting formula is given in [135]). Furthermore, one also must take into account a more complicated mass-radius relation for relativistic SQM stars implied by the equilibrium state numerical solution. For the representative star model of mass $1.4\,M_\odot$ the radius is 10.3 km and the energy to be released scales with the third power of $\chi$ and equals $6.6 \times 10^{54} \chi^3$ erg. For an amplitude $\chi = 0.001$ this would amount to $6.6 \times 10^{45}$ erg in a 20m-wide deposition shell.

The generation rate of radial oscillations of SQM stars was estimated by comparing the timescale of such oscillations with the period of $0.4 \times 10^{-3}$ s, predicted by the model without the excitation energy process accounted for. The timescale is much larger than that for electromagnetic interactions of $10^{-16}$ s. This means instantaneous conversion of the excitation energy into electromagnetic energy inside the SQM star. Within a quarter of the oscillation period, the rate of electromagnetic energy generation is $7.2 \times 10^{49}$ erg/s for $\chi = 10^{-3}$. Further investigations are required to determine how much of the energy will be radiated away and how fast this will proceed.

A rough estimation made in [135] shows that the luminosity for the energy radiated away by the outermost shell of thickness comparable with the photon mean free path would amount to $1.3 \times 10^{34}$ erg/s (for $\chi = 10^{-3}$) with a corresponding effective temperature of $2.0 \times 10^{6}$ K. The total energy $6.6 \times 10^{45}$ erg released in the star would then sustain radiation for $1.6 \times 10^{4}$ years. However, neutrino cooling switching on in $10^{-6}$ s would dramatically cool the star. The estimated neutrino cooling time for the volume of the star, which is $1.4 \times 10^{3}$ s, is $3.5 \times 10^{8}$ times shorter than the electromagnetic radiation time (respectively, for the volume of the initial deposition shell the time would be $1.5 \times 10^{5}$ s, i.e., $3.4 \times 10^{6}$ times shorter than the radiation time).

With the considered excitation mechanism accounted for, the extremely large oscillations, e.g., $\chi = 0.1$, are likely excluded as the energy of such oscillations would be less than the value of the excitation energy generated during the time of a single oscillation, and oscillations would be damped during the first cycle. For lower amplitudes, the energy generation time would be longer. More accurate calculations should take into account the evolution of the temperature of the SQM star; the damping effect on radial oscillations by weak quark processes is rather unlikely to determine the discussed strong interaction phase, although it might dominate the thermalization of excited SQM.

### 4.6. Neutron Star Collapse to Third Family

Massive neutron stars may support in their cores exotic states of matter different from hadronic-like deconfined quark matter. These stars are denoted hybrid stars. In the case of a strong first-order phase transition inside compact stars, their mass-radius relation presents a gap of unstable

configurations between pure hadronic and hybrid stars. The latter ones will populate the so-called "third branch", following the hadronic neutron stars and white dwarfs branches. The transition scenarios may correspond to a neutron star configuration lying at the top end of the hadronic branch whose central density is right below the critical value for deconfinement. There are possible ways for the increase of central density of such a neutron star, thus triggering the transition to a third branch configuration. For instance, a fast rotating neutron star may lose energy by dipole emission resulting in a spin down, or a neutron star in a binary system may undergo an accretion-induced spin-up by matter from a companion. The latter case has been recently proposed as an explanation for eccentric orbits of binary systems where the neutron star is able to accrete matter from a circumbinary disk created after a low-mass X-ray episode. Consequently, the neutrino burst which accompanies the deconfinement phase transition in the neutron star interior may trigger a pulsar kick producing the observed eccentric orbit [136]. A pair of compact stars of about the same mass, each one of them lying in different branches of their mass-radius relation are usually called "twin stars", see [137–139]. The aforementioned dynamical scenarios, where one of the neutron stars collapses into its twin star, may conserve the total baryon number resulting in a mass defect of less than a tenth of the solar mass for state-of-the-art equations of state, which corresponds to an energy of a few $10^{51}$ ergs [140].

It is, therefore, feasible that the possible deconfinement phase transition in compact stars produce energetic emissions, for instance acting as an engine for Gamma-Ray Bursts, with an accompanied characteristic neutrino signal as well as high energy cosmic rays. An analogous situation can be discussed in the framework of neutron star mergers, where both electromagnetic and gravitational radiation are emitted, together with cosmic rays. For the GW170817 event , it has been estimated that the cosmic rays flux is compatible with cosmic-ray detections in terrestrial observatories below the "ankle" [141].

### 4.7. UHECR as the Spacetime Structure Probe?

The CRE hypothesis can be considered an explanation for several outlying cosmic-ray measurements. As an example, in 1975, Fegan et al. observed a simultaneous increase in the cosmic-ray shower rate at two recording stations 250 km apart [142], and in 1981 the cosmic-ray array operated by Smith et al. recorded 32 TeV EAS within 5 minutes, while only one such event would have been expected [143]. Similar cosmic-ray anomalies were not observed again by any of the two groups, nor reported by others. On the other hand, nobody checked further on a global scale. We are going to do so within this project—using the CREDO meta-structure composed of the detectors operated by numerous research groups in all the available energy ranges. This will offer a chance to confirm or question the aforementioned historical reports, potentially leading to the observation of New Physics effects, including probing the spacetime structure. Although the expectations concerning the potential observations of spacetime structure manifestations through effects accumulated over large astrophysical distances are highly uncertain due to missing physics formalism below the Planck scale, as stated e.g., in [144,145], the cumulation of time delays of high-energy photons traveling astrophysical distances is thinkable, especially if one considers photons of different energies emitted simultaneously. The cumulative time delay of higher energy photons with respect to the lower energy ones might or might not depend on photon energies—it is hard to tell due to the missing formalism regarding the proper averaging of spacetime foam fluctuations [144]. (Due to quantum fluctuations, spacetime, when probed at very small scales, will appear very complicated—something akin in complexity to a chaotic turbulent froth, which physicist John Wheeler dubbed quantum foam, also known as spacetime foam.) If the cumulative effects of spacetime foam fluctuations are independent of photon energies, then they might be too small to be observed. On the other hand, if they depend on energy, one cannot exclude time delays even as large as of the order of minutes—comparable to the observations mentioned above. Therefore if, as hoped in CREDO, we can observe CRE composed of high-energy photons of different energies, possibly spanning the whole cosmic-ray energy spectrum, i.e., from GeV to ZeV, then in any case we will have a new "spacetime

probe" at hand—to bring a new input for tuning the existing, or even building the new spacetime models, be it delay evidence or null results that further constrain the available theories. Thus, in any case, we are entitled to expect that within this proposal we will contribute in a novel way to getting closer to understanding the cumulative effects in spacetime foam fluctuations, which would mean a contribution to the foundations of science. Although proper averaging methods of cumulative spacetime foam fluctuations are yet to be developed (as stated in [144]), and despite the uncertainty concerning energy dependence of such effects, not exploring the new research opportunity offered by CREDO would be a methodological mistake. The research in the proposed direction should begin with a wide review of spacetime structure models which predict differences in time delays between the arrival times of high-energy photons of different energies. With such a review the experimental efforts dedicated to CRE could be prioritized accordingly so that appropriate detection and monitoring algorithms could be developed, including the alerting mechanisms on a sub-threshold level to be analyzed in accordance with the multi-messenger astrophysics strategies. The latter direction gives promising perspectives for new advances in astrophysics, reflected also in the new attention of theorists, also those addressing the questions related to space time structure (see e.g., [146–148]) and private interest in discussions (P. Homola's private communication with S. Carlip, Y. J. Ng, and Q. Wang.).

## 5. CRE Simulations

High-energy particles that propagate through the Universe unavoidably interact with background radiation, magnetic fields, and matter, initiating cascades of secondary particles which might be observed with available techniques—like in artificial particle colliders but on a much larger scale. Although an interaction of a particle during its propagation through the Cosmos is commonly approximated with extinction, one has to admit that the logically correct question about chances for observing CRE is not if they exist, but when and how we could observe them. This argument is illustrated with Figure 4.

Let us consider a CRE initiated near the Earth, taking as an example the *preshower effect* [149–152], i.e., a conversion of an ultra-high-energy photon into an $e^+/e^-$ pair due to its interaction with the geomagnetic field above the Earth's atmosphere, and subsequent synchrotron radiation of electrons. The components of the resultant electromagnetic cascade (mostly photons) induce extensive air showers (EAS) upon entering the atmosphere.

Since the distribution of preshower particles above the atmosphere is confined to a very small region (a fraction of $cm^2$) the resultant extensive air shower will have properties very similar to air showers initiated by single primary particles of the energies corresponding to the photon which initiated this preshower. Thus, the observation of a preshower-like CRE will require nothing else but a standard infrastructure dedicated to the detection of UHECR, and in Figure 4 (left panel) we name this scenario an "obvious detection" example.

On the other hand, if a CRE particle distribution is so sparse that only one particle can reach the ideal detector system at a time, there is no technical chance to interpret this individual particle as a component of a CRE. Thus, we consider this scenario an obvious technical limitation for CRE observation, in Figure 4 named "obvious extinction" (right panel). Consequently, the focus of any research dedicated to CRE is limited to the scenarios that can result in particle distributions less sparse than the "obvious extinction" limit, as also illustrated in Figure 4 with the "obvious between" CRE case (middle panel), so far unexplored with a globally coordinated observational effort.

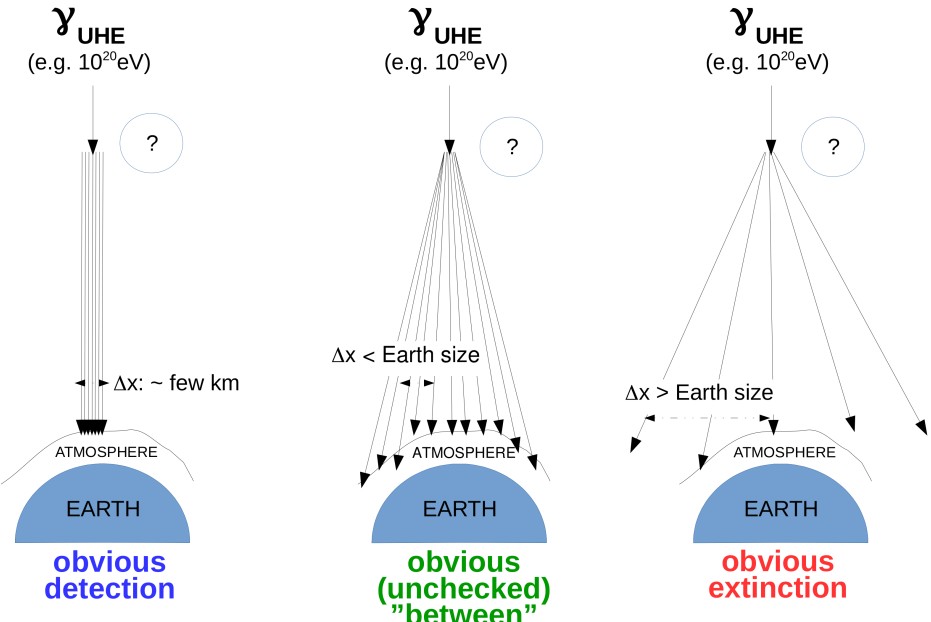

**Figure 4.** Obviously in experimental reach, although not yet probed, detection of Cosmic-Ray Ensembles, using ultra-high-energy photons as example primary particles.

As far as CRE scenarios are concerned, the ongoing simulation studies include synchrotron radiation of high-energy electrons in the presence of planetary, stellar and galactic magnetic fields. As shown, for example in [41], in the case of the preshower effect occurring due to UHE photon interactions with the geomagnetic field, the resultant EAS can hardly be distinguished from those initiated by individual particles based on standard air shower observables like the atmospheric depth of shower maximum development or muon content in the particle distribution on the ground—unless a large statistics of events is available. Given the stringent UHE photon limits one does not expect many photon- or preshower-induced events to be observed even with the largest air shower detector arrays such as the Pierre Auger Observatory or Telescope Array. Instead, one might consider alternative observables and alternative infrastructures, more sensitive to electromagnetic multi-primary EAS origins, e.g., those connected with Cherenkov emission induced by air shower particles and being observed by gamma-ray telescopes. A study in this direction was presented in Ref. [153], where the authors analyze the feasibility of detecting preshower-induced EAS using Monte Carlo simulations of nearly horizontal air showers for the example of the La Palma site of the Cherenkov Telescope Array (CTA), as illustrated in Figure 5.

It was demonstrated that there is a realistic chance for identification of preshowers induced by 40 EeV photons coming from an astrophysical point source during 30 hours of observation. This result confirms that the Imaging Atmospheric Cherenkov Telescope technique, although normally dedicated to the observation of GeV-TeV gamma rays, can also be used to observe phenomena occurring in the UHE domain, provided that an efficient photon/hadron separation is obtained. The small expected rate of preshower events is a direct consequence of the small flux of UHE photons predicted by both top-down models and by the GZK effect.

Nevertheless, in [153], transient events such as gamma-ray bursts (GRB) are considered to be potential sources of UHE photons. By comparing the gamma-ray flux at lower energies before and during the GRB, and after extrapolating the ratio to UHE domain, the expected UHE photon fluxes for transient sources were obtained. In the case of MAGIC's observation of GRB 190114C [154], such *boosting* of the gamma-ray emission in non-transient mode can reach a factor of up to 652, potentially producing an UHE photon flux significantly higher than upper limits put by the Pierre Auger Observatory or Telescope Array.

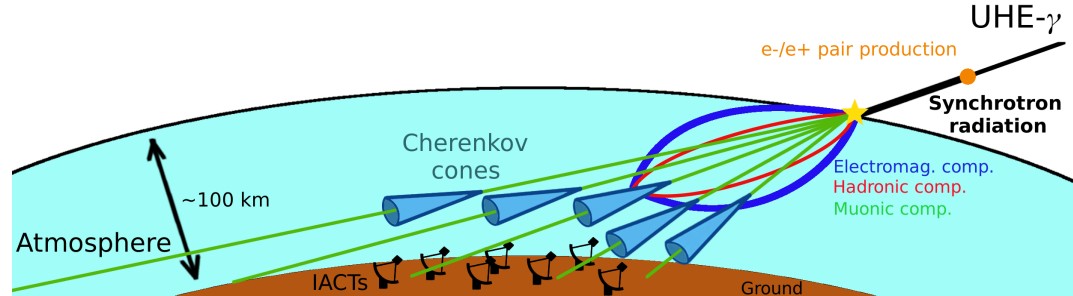

**Figure 5.** Schematic view of the preshower effect observed by gamma-ray telescopes; in the geomagnetic field, an ultra-high-energy photon is converted into an $e^+/e^-$ pair, which radiate synchrotron photons. A CRE composed of thousands of photons and a few electrons/positrons reaches the top of the atmosphere within a few square centimeters. An EAS is subsequently produced and, due to the high zenith angle observation mode, the electromagnetic and hadronic components are mostly absorbed. The surviving muonic component emits Cherenkov radiation which can be detected by ground-based Imaging Cherenkov Telescopes [153].

In this context, increasing the probability to observe the preshower effect with gamma-ray telescopes could be obtained through multi-messenger alerts adapted to the nearly horizontal observation mode. Such a strategy has already been adopted by existing astroparticle physics experiments and allowed the observation of a correlation between neutrino emission and the gamma-ray emitting blazar, TXS 0506-056 [155]. Therefore, a campaign of observations of high-energy sources with gamma-ray telescopes, for an observation time far greater than 30 hours, constitutes a great opportunity to detect a signal yet unseen. Consequently, the connection between CREs and gamma-ray telescopes seems evident and an effort should be made to include gamma-ray astronomy in CREDO's strategies.

To generalize the notion "preshower", the term "*super-preshower*" was introduced (see e.g., [152]). Super-preshowers (SPS) are cascades of electromagnetic particles, thus a subset of a potentially wider CRE family, initiated above the Earth's atmosphere in a process that occurred at an arbitrary distance to the observer - not necessarily within the geomagnetic field. Super-preshowers can be classified with respect to their spread in space and time, potentially the most promising observable properties of CRE. For an example, a cascade initiated due to the preshower effect in the geomagnetic field (even at altitudes as high as 10000 km a.s.l.) is expected to have the lateral spread above the atmosphere (100 km a.s.l.) in the order of millimeters, and negligible spread of arrival times [151,156]. If the preshower effect was to occur in the vicinity of the Sun, one would expect still negligible spread of arrival times, but the lateral spread then might reach the size of the Earth [152] or even larger. The resultant SPS signature is then expected to be composed of even hundreds of thousands of extensive air showers, forming a characteristic, very thin (order of centimeters) and elongated (up to millions of kilometers) pattern.

This example shows that by analyzing the properties of CRE/SPS one might approach attributing a non-trivial physical scenario to the observable event category, provided the underlying uncertainties are properly understood and quantified—as planned by CREDO. The few calculations performed so far in this direction can still serve only as qualitative indications concerning potential CRE/SPS observables. For instance, in Ref. [157] the lateral spread of an SPS originated nearby the Sun is simulated using a private code, assuming that an SPS is composed only of photons of energies larger than E = $10^{17}$eV. A more detailed calculation reported in Ref. [152] and obtained using an open source public code (PRESHOWER [151,156]) shows that the energy spectrum of SPS particles might extend down to even quite low energies, with a peak at around 10 GeV, as illustrated in Figure 6. TeV photons would certainly induce air showers that would contain particles, mostly muons, observable on the Earth's surface. While in large cosmic-ray observatories these muons are treated as background, the complete CRE/SPS-oriented research we propose here has to include a proper handling of this

"unwanted muon background" which can be processed to extract a signal induced by ensembles of low-energy air showers arriving simultaneously at the detector—very clear, unique, and so far untested signature. We make one step further and propose a global analysis of the data from the available detectors to search for extremely spread CRE/SPS events, inaccessible by the largest observatories taken individually. The experimental question which can be addressed in this regard is: "which CRE/SPS fronts, and in which circumstances, can be detected by a network of devices located on Earth or around it?". It is the question about the SPS/CRE particle density on the top of the atmosphere (or speaking more general: on top and within the technosphere) and the particle density, or any other observable information, such as e.g., Cherenkov light, related to the resultant air shower ensembles.

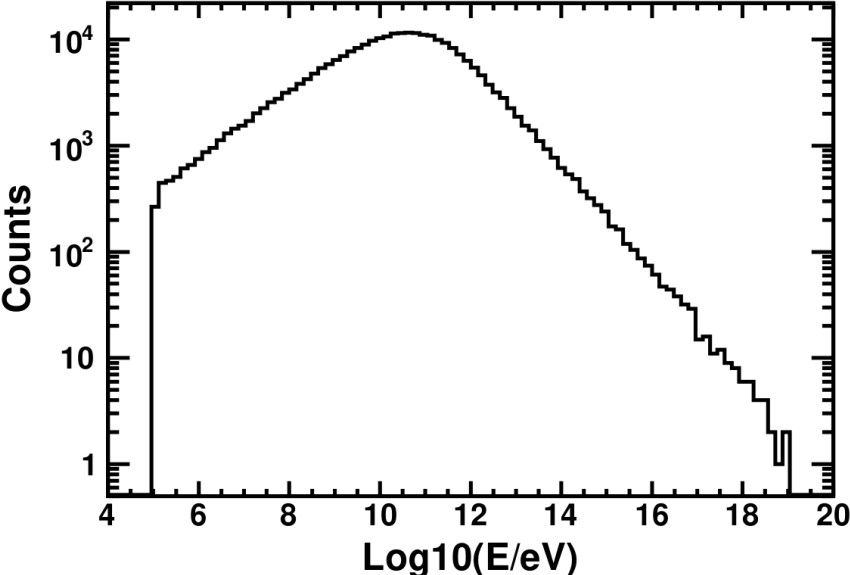

**Figure 6.** An example energy spectrum of a cosmic-ray ensemble generated by an ultra-high-energy photon in the vicinity of the Sun, after the interaction with the solar magnetic field (primary photon energy: $10^{20}$ eV, the heliocentric latitude of "the closest approach point": $0^o$, and the impact parameter: $3\,\mathrm{R_\odot}$) [152].

Another type of a CRE scenario, where electromagnetic particles play a role, is the propagation of very-high-energy electrons through intergalactic and galactic magnetic fields. One of the analysis in this direction currently being carried out within the CREDO Collaboration concerns the propagation of electrons of energies between $10^{17}$ eV and $10^{19}$ eV within the Galaxy, using CRPropa [158]—a Monte Carlo simulation of cosmic-ray propagation and the state-of-the art modeling of the galactic magnetic fields to quantify the chances of observing a CRE on Earth. The intermediate results indicate qualitatively that one might have a chance of observing a CRE originating from synchrotron radiation occurring within the Galaxy or maybe even at some other extragalactic sources. Quantification of the observational chances is still on the way, but already the qualitative study conveys a very important message: even "conventional" and abundant electromagnetic processes such as synchrotron radiation in galactic magnetic fields are expected to generate CRE reaching the Earth.

At the end of this section it is worth emphasizing that the experimental strategies dedicated to CRE do not need to rely on specific theoretical scenarios—one might also "fish" for clearly non-random cosmic-ray global footprints. This approach is justified by the ultra-high-energy physics uncertainties we are aware of—large enough to admit that we might not be able to imagine *all* the possible physics scenarios predicting a CRE signal that would be observable on Earth. It is, therefore, sensible "just" to explore *fully* the potential of the infrastructure we have at hand and go fishing for signatures we are not able to predict, but which we can distinguish from the diffuse (random) cosmic-ray background.

## 6. CREDO Detectors: Cloud of Clouds

As explained above, any experimental strategy oriented on observation and investigation on large scale cosmic-ray correlations in the form of widely defined CRE require a global approach to cosmic-ray research. Since both extensive air showers and incoherent cosmic rays might originate from a CRE, one realizes that in fact any detectors capable of detecting cosmic-ray signals, both on the ground or on satellites, can potentially serve as valuable components of a global data acquisition system. In other words, the chances for CRE observation increase with any single detector or observatory joining the collective observational effort. Within this general concept, schematically depicted in Figure 7 the role of CREDO can be understood as an umbrella research program that enables a collaborative effort dedicated to CRE with the use of existing infrastructure and expertise, and with openness for designing and building complementary detectors or arrays—if required and justified by specific CRE models and research plans.

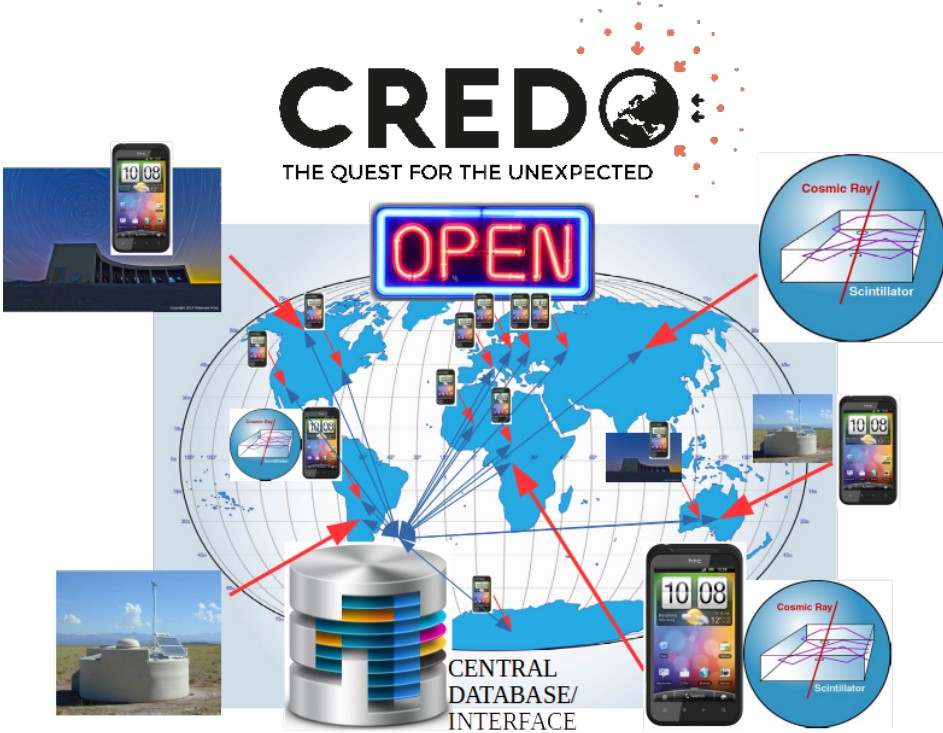

**Figure 7.** The concept of the Cosmic-Ray Extremely Distributed Observatory (CREDO): open on two ends (data upload and access), using both professional, dedicated cosmic-ray infrastructure and off-the-shelf detection solutions, including smartphones.

Widening the participation in the CREDO program increases its scientific potential. The CREDO program benefits all participants by being as open as possible through the following: removing or reducing as many non-scientific barriers that block findability, availability and interoperability of contributing data sets as possible, and facilitating the usage of the systems, taking into account different needs and levels of expertise of the individual participants and institutional stakeholders. To illustrate technological diversity behind the general CREDO concept in this section, we briefly sketch a landscape of detection techniques used in the state-of-the-art cosmic-ray research with particular emphasis on the current situation, needs and interests of the CREDO Collaboration.

### 6.1. Cosmic-Ray Detection Techniques

The current CREDO collaboration with existing experiments and all the future planned detector installations allows for the extension of the widely used EAS (Extensive Air Shower) detection and

analysis techniques. EAS detector systems are typically designed as a grid of individual detectors that spreads over a large area. This is due to two reasons: the disk of the EAS at the observation level can be up to a few kilometers in diameter for ultra-high-energy events, and the frequency of such events per $km^2$ is low so higher grid area means higher event rate.

The most common technique that is applied as part of data analysis uses the particle density distribution within the EAS disk. The average distribution of the particle density in the disk varies with the distance from the shower axis in a known way; it additionally depends on the energy of the primary – $E_0$. Figure 8 [159] illustrates the distribution of the particle densities in simulated EAS disks at the different distance from the axis of simulated showers. This is shown for several values of the $E_0$ of the primary protons. The simulations were done using CORSIKA [160] software package. Please note that the particle density distribution is non-linear (approx. inverse quadratic far from the axis). Additionally, each individual detector within the grid provides the time of the hit, e.g., some timing information when the signal in this detector went over a preset threshold. This information is used to determine the EAS arrival direction.

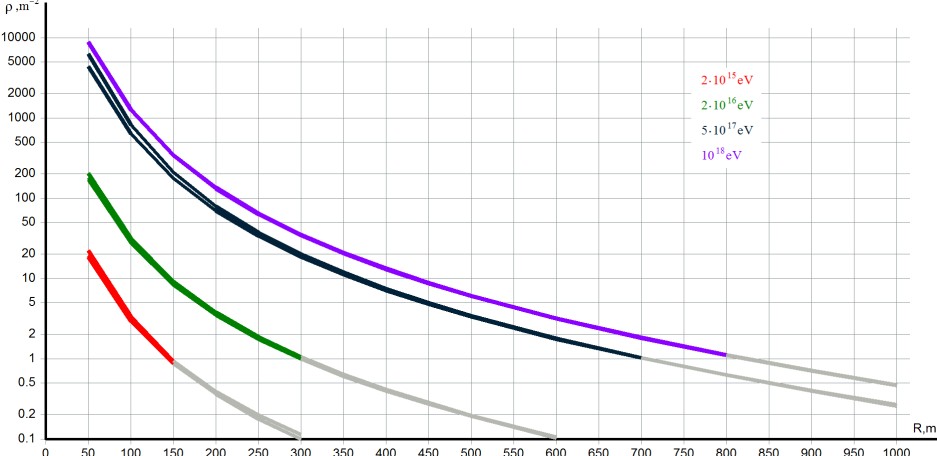

**Figure 8.** Particle density distribution in simulated EAS disk vs. distance from axis for different $E_0$ [159].

However, there is an unused piece of information here. As the EAS disk passes the detection level, the detectors can measure not only the integral number of detected particles but their distribution over time as well, getting an effective width of the disk [161]. This technique was first developed and used at the Horizon-T [162] experiment that holds close ties with the CREDO collaboration. This approach uses the width of the EAS disk as measured by different detectors during the event. Figure 9 [159] illustrates the widths of the same simulated EAS from Figure 8. The grey lines of the plot show approximated widths at the distances from the axis where particle density is low for reliable determination. The width behaves in a more linear way when compared to particle density, specifically closer to the axis, and is weakly dependent on the primary particle energy $E_0$. Using all available information from arrival time , particle density distribution and disk width, one gains advantages such as the ability to do some analysis on the events with axis outside of the detector active area. This ability is due to the fact that the approximate position of the axis position can be reasonably estimated from the disk width information. However, such detectors and specifically electronics are more costly as fast Flash ADC and fast PMT with matching detection medium are required [163].

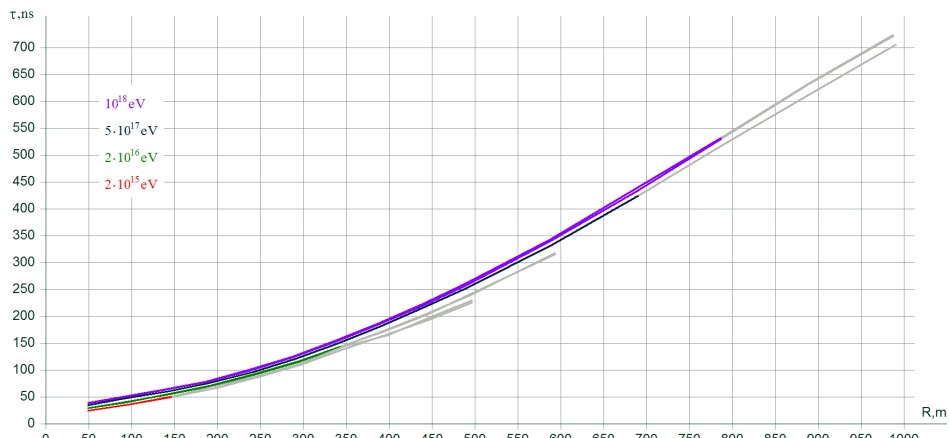

**Figure 9.** Simulated EAS disk width vs. distance from axis for different $E_0$ [159].

*6.2. Overview of the Different EAS Detection Techniques—Scintillator, Water Cherenkov, CMOS/CCD, Air Fluorescence, Radio*

Numerous methods are used by existing experiments to detect charged particles from the EAS disk. As we cannot see the particles themselves, the principle is to convert the passage of the particle through the detector into a signal that is easy to process, or observe the result of the particle interaction with the detection medium.

### 6.2.1. CMOS/CCD

The most direct method involves using a silicon-based pixelated strip. The most commonly found detectors of this type are in everyone's cell phones and photo cameras – they are the CMOS/CCD photosensors (more information is available in [164]). On passage through the pixel, charged particles as well as gamma and x-rays affect CMOS/CCD sensors in a similar way as light does. If the camera cap is closed (or cell phone lens is well covered) so that the sensor is in the dark, the pixels affected, or hit, by a particle passage will produce a response as if they were exposed to light. If more than a single pixel is hit, a part of the particle track may be detected as well as shown in Figure 10. This method is currently used by the CREDO Collaboration using the cell phone cameras and a special app [7,8] extendable or portable to a browser application.

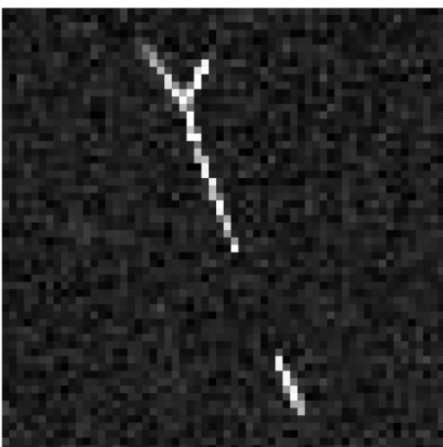

**Figure 10.** A sample event of a cosmic ray leaving a track on a CCD sensor [8].

### 6.2.2. Water-Based Cherenkov Detector

Another detection method that is based on observing the results of particle-matter interaction relies on the detection of the Cherenkov light – an electromagnetic shock wave from the charged

particle passing the transparent medium (water, air, glass, plastic) faster than the speed of light inside that medium (read more on Cherenkov radiation in [165]). The shock wave consists of thousands of photons per 1 cm of particle path in the medium that can be detected using fast photodetectors with internal signal amplification, such as PMT (Photo Multiplier Tube).

The design of the water-based Cherenkov detector is very simple – an insulated volume of water and at least one PMT that is optically connected with this volume. A charged particle creates a cone of Cherenkov light that is detected by a PMT. If the volume is very large and many PMTs are used, the Cherenkov light cone can be imaged and used for measuring particle momentum and direction: this technique is used, for example, by Super-Kamiokande detector (more information please see [166]).

The advantages of water-based detectors are the detection speed (on the order of one nanosecond [163]) and relatively lower cost since the detection medium is water (typically, ultra-pure). However, the Cherenkov signal is relatively weak. Other disadvantages are low to no mobility and the insulation of high voltage electric components like PMT from the water.

### 6.2.3. Scintillator-Based Particle Detectors

The scintillators are the materials that produce a so-called scintillation light in response to a charged particle passage in addition to Cherenkov light. The main advantages of scintillators over water or glass as detection medium is that there is about 10–15 times more scintillation light photons than from Cherenkov radiation. Also, most scintillators are solid and the detector is easily movable in most cases and does not require a lot of additional external support, but there are liquid and even water-based scintillators [167]. Specifically, plastic scintillators are lightweight and inert solids that can be used for production of detectors that are suitable for installation at schools. In such cases, PMT cannot be used due to high voltage being a hazard, but there are silicon versions of PMT that work at the safe voltage such as SiPM, MPPC, MRS etc. [168,169]. Thus, scintillator-based detectors are most suited for CREDO long-term outreach and educational goals. The main disadvantages are the cost and a slower response when compared to Cherenkov light [170]—on the order of 10 ns for plastic and liquid scintillators, on the order of 100 ns for inorganic.

### 6.2.4. Air Fluorescence Detectors

As the EAS disk travels through the air, the charged particles that comprise it ionize the air molecules along the entire EAS path. As the electrons return to their ground state, the energy is released as light. This light can be collected by typically the collection of mirrors with multiple PMTs as light detectors (there are various designs and detectors are used). The main advantage for this method is that with some luck when viewed at the right angle, an actual development of EAS can be captured and accurate estimates of the properties of the parent particle can be made. The main con is that the observations can be done only during Moonless and clear nights. Fluorescent detectors are often combined with scintillator or water Cherenkov detectors. More information can be found in [171].

### 6.2.5. Radio Signal Detectors

As the EAS disk moves in the Earth magnetic field, the positive and negative charges within it experience Lorentz force in opposite directions and radiate in a radio frequency range around 30–80 MHz or so [172]. This effect is highest for the EAS closer to the horizon and moving perpendicular to the magnetic lines of Earth. Although many experiments are actively trying to use this method, it is a supplement to all other detection methods listed.

### *6.3. The CREDO Extension Proposals*

The involvement of non-scientists and outreach is one of the pillars for CREDO collaboration. There are currently proposals to design and produce portable, most likely scintillator-based EAS detectors that could be used at education establishments (schools, universities) as both demonstrators

and part of the education process. The designs are all being proposed with the flash ADC capabilities to use this expanded set of analysis techniques that involve using advanced EAS timing information and possibly extend the searches for new phenomena within EAS such as 'unusual' events described in [173,174].

One of such designs is CREDO-Maze, a project that will create a global, unique physical apparatus, which will consist of a network of local (school) measuring stations. The idea of the CREDO-Maze project is based on the Roland Maze Project [175,176] developed 20 years ago. Today's technology has evolved considerably, and it is now much easier and, critically, much cheaper to implement the local shower array idea of Linsley [177]. It is our aim to provide high schools with sets of at least four professional small cosmic-ray detectors linked locally and constituting a school EAS array. The feasibility of EAS detection with such a mini-network was demonstrated in Ref. [178] where pocket-size (sensitive surface of $\sim$ (25 cm$^2$) affordable (cost $\sim$100\$/piece) scintillator detectors [10] were used. The CREDO-Maze project uses technologically sophisticated measuring equipment in extracurricular activities: detectors of charged relativistic elementary particles will be made of small (0.02 m$^2$) plastic scintillators. The light pulses will be collected by optical fibers shifting the wavelength from ultraviolet to green and then light will be converted into electrical signals by Silicon Photomultipliers. Further electronics will be based on high speed digital circuits and microcontrollers to connect to higher-level servers via the Internet and WiFi links. Prototypes of individual components of the apparatus have been largely developed independently by several institutions.

One of the important parameters of proposed equipment is the cost. It is easy to build expensive and complicated, 100% effective professional arrays. We are on the way of building the prototype which cost (including scintillators, SiPMs, trigger electronics, storage and data transmission micro-computer) is below 200 \$ (compared to 3000 € per detector for $\mu$Cosmics detector in [179]. Prototypes of individual elements of the apparatus have been largely developed independently in several partner academic centers: University of Lodz, National Centre for Nuclear Studies and Institute of Nuclear Physics in Poland, Institute of Experimental and Applied Physics, in Czech Republic, Swinburne University of Technology in Australia. Completion of the whole and its technical adaptation for replication will be one of the interesting tasks of the project. It is an interesting concept to deliver to some of the end user (school) kits, which are adapted for this purpose, assembled only in basic, skill-intensive parts. This would allow students in their local project groups to build and assemble from them a fully operational and efficient whole, under the supervision, of course, of the staff of the institutes managing the local project networks. The independent construction of the operating scientific equipment is an additional motivating element and undoubtedly increases the involvement of young people and the general interest of those not participating in the project. These effects were observed in previous attempts to implement similar activities on a smaller scale.

On the other side it should be mentioned that proposed devices are designed and implemented in such a way that while maintaining high standards, they are as inexpensive as possible. Technologies will be developed to ensure that the measurement kits can be duplicated and distributed to end users as "self-assembly kits" with different degrees of sophistication of the finished components. As potential business projects they will be able, together with educational material pledges and software, to provide a ready-made market product. With positive recommendations based on our research results, the potential market, the demand of educational institutions, seems to be quite considerable. The creation of local structures comprising young people involved and organized in research groups (led by teachers/educators) using network communication and based on science centers, as, e.g., higher education institutions, universities, is an important step in the development and institutional activities research performing organizations, including , as well as research funding organizations. The proposed actions open up new areas of innovation in non-formal non-school education. Creating a model system of social communication networks and demonstrating its effectiveness in the proposed field being an element of STEM will allow planning and creation of similar networks realized in other areas of

education. There are no contraindications for such networks to cover various groups of young people and research centers.

Another concept of building a large scale cosmic-ray network is based on engaging the wide community, thanks to the attractive properties and portability of devices such as pixel cameras. A benchmark standard here is Minipix Timepix EDU, a hybrid semiconductor pixel detector, with a silicone sensor of various thicknesses (e.g., 300, 500 µm) provided by ADVACAM [180], using the technology developed within the Medipix Collaboration at CERN [181]. This detector provides specific characteristics: fast-data acquisition, portable, lightweight, easy to use and easy to place in difficult-to-access environments, and can be operated remotely. Minipix Timepix EDU is a simplified and price effective version of the Minipix Timepix detector. It was designed and created with the purpose to make science accessible to the public (schools, scientific research centers, non-commercial institutions). The simplified version of the software Pixet, with predefined settings, makes it possible to operate the detector without advanced training in data acquisition just by connecting the device to the USB port of the PC.

The ASIC read–out chip contains a matrix of 256 by 256 pixels (total 65,536 independent channels) and an active sensor area of 14 mm by 14 mm, where one pixel corresponds to 55 µm, see Figure 11. The dark noise signal enables measurements of low–Linear Energy Transfer (LET) particles with high precision. Given to the per-pixel calibration, an adjustable threshold is made in each pixel. This allows a detection efficiency close to 100% for heavy charged particles, making these detectors unique and suitable for the detection of cosmic rays.

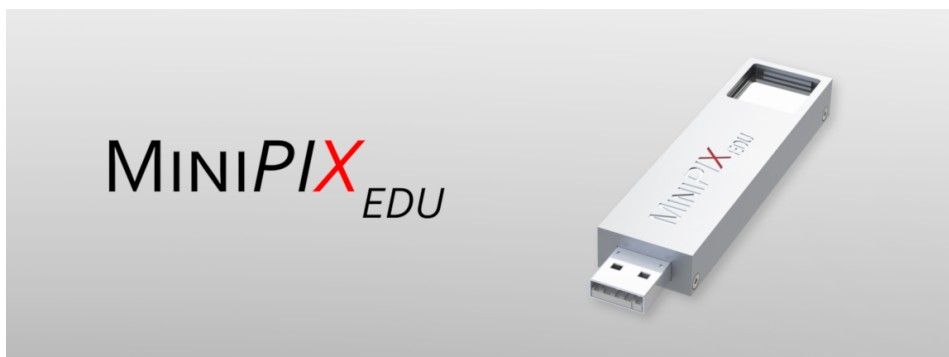

**Figure 11.** Illustration of the Minipix Timepix EDU detector dedicated for educational purposes, www.advacam.com, accessed on 22.09.2020.

Online Track Visualization and Processing Software

The software package Pixet Pro (D. Turecek, J. Jakubek, 2020, Advacam s.r.o., Prague, Czech Republic) is used to operate the detector and to control the readout, data acquisition and recording. The detectors can be connected to a standard notebook using a USB 2.0 port. The PC connectivity and cross-platform operating system compatibility includes Windows, Linux and Mac OS.

An example of a mixed field frame can be seen in Figure 12. Large roundish blobs are created by alpha particles, long strikes by cosmic muons, curving tracks by electrons or small dots by gamma or X-rays. Moreover, rare and exotic events can be observed: Delta electrons, recoiled nuclei, cascade of two or more nuclear transitions, proton tracks. According to the frame occupancy the acquisition time can be set. Other parameters such as threshold are already predefined. Furthermore, the data can be processed as dose rates, absorbed dose, fluence maps, energy deposited, LET spectra [182].

A notable role in the implementation of the CRE-oriented detection strategies is also going to be played by professional, medium size detectors giving a high-quality signal.

Since 2018, low-background, digital gamma gamma-ray spectrometer with active shield has been operating in a ground level laboratory in the Department of Nuclear Physical Chemistry, Institute of Nuclear Physics Polish Academy of Sciences (IFJ PAN), Krakow, Poland. The spectrometer is equipped

with Broad Energy Germanium detector BE5030 (Canberra, USA), multi-layer passive shield and five large, plastic scintillators (Scionix, Netherlands), playing a role of additional active shield (veto system). Areas of scintillation detectors range from ca. 0.14 up to 0.49 m$^2$. Data acquisition as well as signal processing are performed by means of digital analyzer (so-called digitizer) DT5725 (CAEN, Italy) [183].

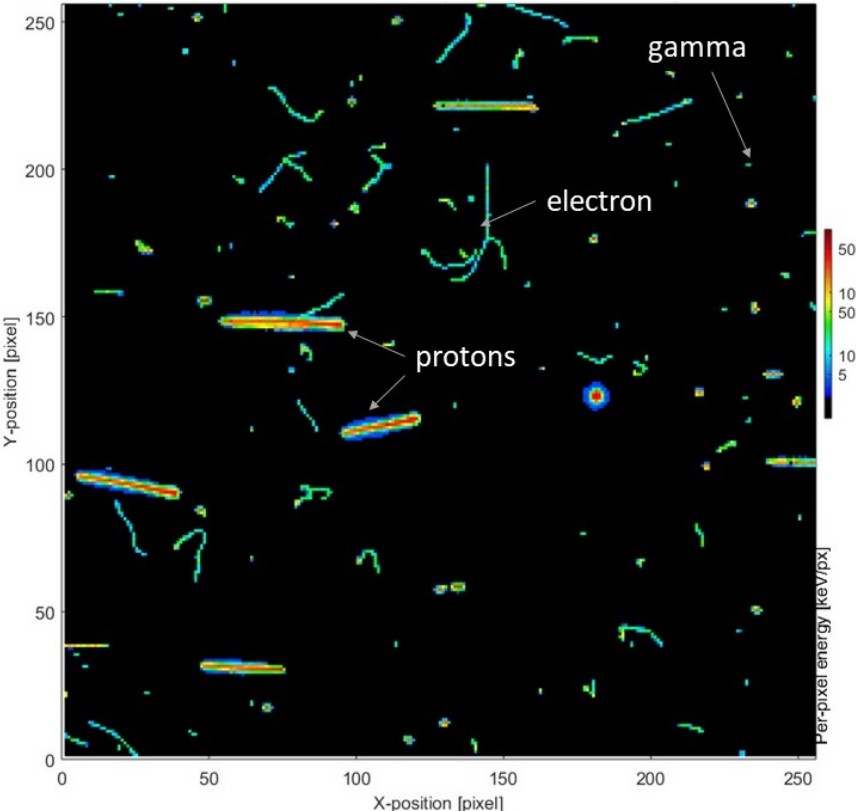

**Figure 12.** An example of per-pixel energy deposited by various particles in a mixed radiation field measured using a Minipix Timepix detector with Silicone sensor. The detector was operated in ToT-frame mode with an acquisition time of 10 ms. Low-energy, narrow, curly tracks are typical for electrons, high-energy, wide, straight tracks for energetic heavy charged particles such as protons.

The main role of both passive and active shielding is reduction of the radiation background of the germanium detector. However, thanks to digital data acquisition it has become possible to expand the research potential of the constructed spectrometer. Registering and storing data generated by all spectrometer's detectors and using manifold, off-line data exploration techniques allowed one to initiate continuous monitoring of the cosmic-ray muon flux.

From the CREDO Project point of view, such device may be used as a *reference detector* because scintillators register several dozen of muons per second and thanks to their non-collinear orientation (Figure 1 in [183]) it is possible to detect at least 3 muons correlated both geometrically and in time. Preliminary research showed that during single gamma-ray spectrometry measurement that lasts 426721 s (about 5 days), scintillators registered 329 events of 5-fold detection coincidences of muons (Figure 13) that could originate from air showers.

Additional advantage of continuous monitoring of cosmic-ray muons flux is the possibility to investigate correlation of changes in its intensity and Earthquakes that modulate the local geomagnetic field [184].

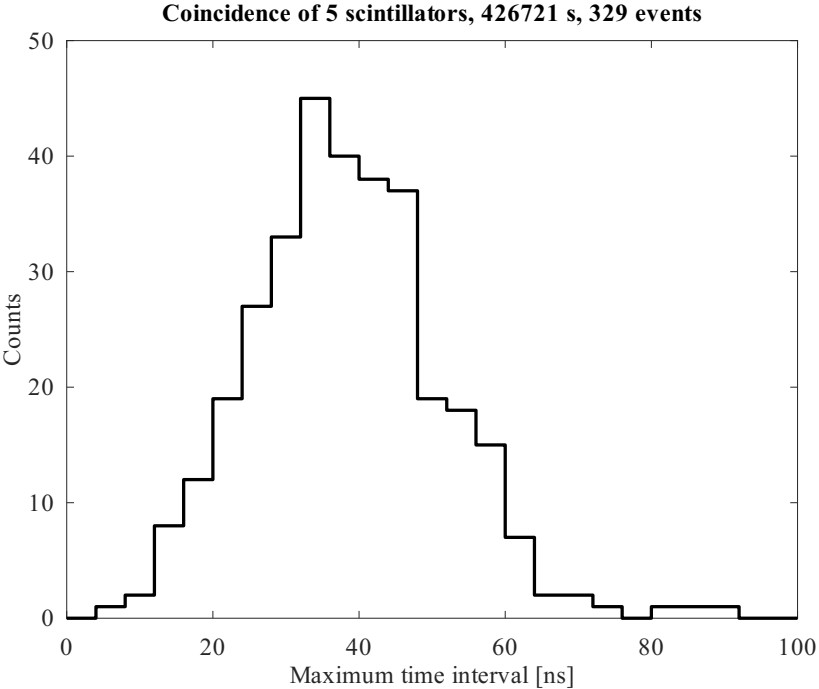

**Figure 13.** Distribution of maximum time intervals of 329 events of five-fold coincidences registered by spectrometer's active shield during single gamma-ray spectrometry measurement (time of measurement 426721 s).

*6.4. Inter-Detector Communication*

An important aspect of R&D concerning the CRE detection feasibility is communication between detectors constituting an array. It requires attention especially in deployment areas with limited access to internet and/or electricity, where data received by an individual detector cannot be transferred directly to the central acquisition system. Communication might also be critically important in situations which require some preprocessing of the data collected by a subset of the array, before sending a trigger message to the central system. The communication issues concerning the CRE-related applications were addressed in Ref. [185]. They are the subject of further ongoing investigations and engineering works, below we briefly summarize the current status of these efforts.

We focus on a scenario, when designing the network, which assumes that the detectors are mobile devices that can be located in hard-to-reach areas, such as deserts or forests, as in highly urbanized areas, i.e., cities, and even inside blocks (Figure 14). Such assumptions force the network to have the following features:

- scalability—it is easy to connect new devices to the network, the network should be self-configuring
- low energy consumption during data transmission—an indispensable parameter to ensure support for mobile devices (without access to power from the network)
- universality—the network must operate both in dense urban buildings and in desert or forest areas
- wireless—provides the ability to collect data without the need for expensive infrastructure in the form of cables and the need to use human force (manual data collection from SD cards), in other words, significantly reduces the cost of maintaining the project
- as long range as possible—guarantees stable transmission without the need to use re-transmitters or uneconomical use of too many gates

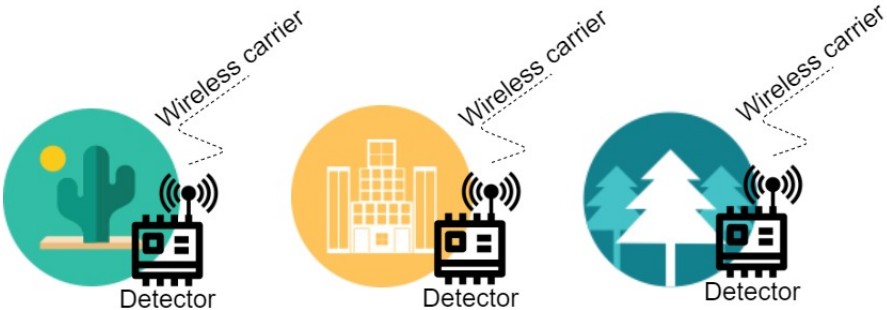

**Figure 14.** Symbolic diagram for the CREDO network scenario.

6.4.1. Solutions Available on the Market in Wireless Networks

To choose the right solution for wireless transmission in the CREDO network, the solutions already existing on the market were compared in terms of parameters important for the CREDO project. The results are presented in Table 1. The table completely omits Bluetooth technology due to the very small range (10–50 m) [186], which does not meet the long-range criterion.

**Table 1.** Comparison of wireless communication solutions available on the market.

| Feature | ZigBee [187] | LoRa [188] | SigFox [189] | WiFi [190] | GPRS [191] |
|---|---|---|---|---|---|
| **Frequency** | 868/915 MHz and 2.4 GHz | 100 MHz to 1.67 GHz | 868/915 MHz | 2.4 GHz | 900–1800 MHz |
| **Power consumption Tx** | 37 mW | 100 mW | 122 mW | 835 mW | 560 mW |
| **Range** | 100 m | 5 km | 10 km | 100 m | 1–10 km |
| **Cons** | Requires infrastructure | Available Gateways on the market are only for 438 and 868MHz | Requires infrastructure similar to GSM (masts and receiving stations) | No Internet connection in non-urbanized areas; high power consumption | No access in non-urbanized areas; high power consumption |
| **Suitable for battery devices?** | Yes | Yes | Yes | No, too much power consumption | No, too much power consumption |

Based on the table above, ZigBee, LoRa and SigFox were selected for testing and deeper analysis. It is worth noting that the ranges given in the table are the best result, often achievable only in an open area, not in built-up areas. It was decided that all tests will be carried out in built-up areas, considering city centers as the most problematic environment for establishing long-range wireless communication. No re-transmitters were used in the tests. The first tests were conducted for the ZigBee network. We managed to achieve a range of about 30 m, similar to that presented by Kuzminykh et al. [192]. Then, tests were carried out for the sigFox network [193]. Due to the necessity to use ready-made SigFox infrastructure and the still incomplete coverage of the area with this infrastructure, further development work using this standard was abandoned. Finally, tests were carried out on LoRa modules [193,194]. We did not manage to achieve satisfactory results (100–300 m). Experiments carried out by Centenaro et al. [195] in built-up areas in cities that used the actual LoRa network and its ready infrastructure also did not reach the maximum range of 5 km.

After the tests, it became clear to us that in order to obtain good coverage both in the city without the use of repeaters, and in open areas, where there is no network infrastructure. We need to develop our own system. We immediately decided to work at lower frequencies (169 MHz) so as to minimize the attenuation of waves by obstacles [196].

6.4.2. Definition of CREDO Wireless Sensor Detector Network

The CREDO Wireless Sensor Detector Network (CREDO WSDN) is a fusion of the well-known Wireless Sensor Network (WSN) currently used most frequently in solutions for the Internet of Things (IoT).

CREDO WSDN is a wireless network, organized in a star topology, based on radio waves at the frequency of 169MHz, used to transmit information to the Sink collective station, from dedicated mobile cosmic-ray detectors. Sink has a COM connection to the PC station, which in turn is connected to the Internet. The internet provides the end user (e.g., a scientist) with access to the collected data. Additionally, additional sensors, such as humidity or temperature sensors, can be connected to the CREDO WSDN to investigate the correlation between events detected by the detectors and other parameters in the area. Figure 15 presents the CREDO WSDN topology.

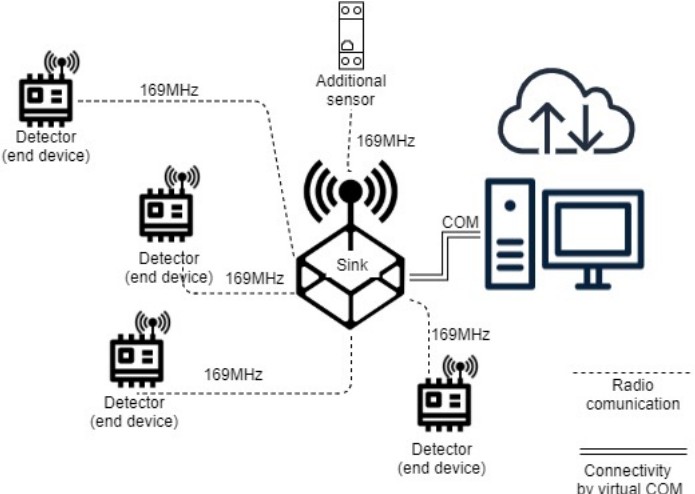

**Figure 15.** CREDO WSDN topology.

Due to the fact that the network is to be especially adapted to communication on mobile devices (mobile detectors), in the current version of the system communication is only one-way, i.e., the data from the detector is sent to Sink. The radio transmitter will be integrated with the detector itself, not a separate device as it was in the first version of the system. These measures are primarily intended to reduce the power consumption of the detector itself, which is to operate on battery power.

The CREDO network consists of several main elements, such as radio communication on 169 MHz, a Sink station, a microcontroller or a power source. The features and role of the most important elements of the entire system are discussed below.

Radio Communication

Radio communication takes place at 169 MHz, unidirectional in a star topology. These features ensure not only excellent coverage in built-up areas or in buildings, but also the optimization of energy consumption—only the Sink station must remain listening all the time. At the moment, the best result that we have achieved is a range of 8 km in built-up areas.

Transmitter

We plan to introduce a new version of the transmitter. This year, a new STM32WL chip is to appear on the market—it is a microcontroller integrated with the radio [197]. It has excellent sensitivity and transmitter power. At the same time, the energy consumption is half less of the previous solution, thanks to the use of a built-in impulse converter. Usually, such a solution significantly worsens the sensitivity of the receiver, but this time it does not have this effect. Additional feature will be that the same system will count pulses caused by particle impacts.

Receiving Station (Sink)

The receiving station in current version already has very good sensitivity [185] however, as we plan to upgrade the transmitters, there is a prototype of a new receiving station as well, currently during testing. The new version with a new LNA (Low Noise Amplifiers) amplifier with lower noise and much lower distortion for now increased the range by 4 km compared to the previous version (from 4 km to 8 km). A new version will be tested as well with several antennas to increase the sensitivity.

Mobile Detectors

The reason the network was created was the need to handle the transmission between battery-powered devices, and more specifically mobile cosmic-ray detectors. At the moment, the first prototype of the scintillation detector has been developed, the tests are showing promising results, and the device is already adapted to battery power.

Microcontroller

The microcontroller acts as the brain of the entire system, makes decisions about how to configure the network, when to transmit the collected data, and informs Sink about the transmitter battery status. It is found both in Sink, transmitting stations and in the detector.

### 6.4.3. CREDO Network Tests in the Field

During the tests on the first version, it was possible to achieve a satisfactory range of 4 km in built-up areas [185]. Currently, after changes to the design of the transmitter and the collection station (Sink), the range has increased to 8 km. Also, there was significant improvement in range in Elevated Areas (mountains). In the new version of the receiving station, we managed to establish communication on a cloudy day, in the area where the mountain was located. An interesting phenomenon that we were able to observe was the reflection of the wave from the stratosphere on a cloudy day (Figure 16) [196].

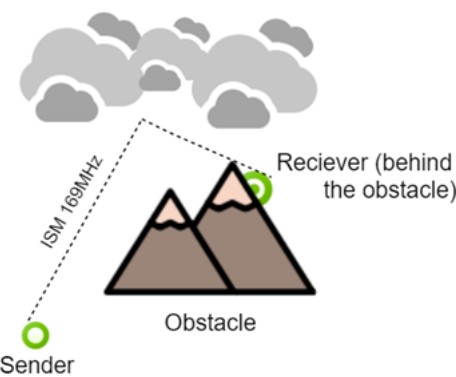

**Figure 16.** Reflection of the radio wave [196].

### 6.4.4. Future Work and Conclusions

The CREDO communication system, thanks to its scalability and universality, can be used in many areas of IoT and can be an alternative to existing solutions, where due to the lack of infrastructure or too high costs of their implementation, they cannot be launched. There are still many challenges ahead of us, we are constantly working on increasing the range. Maintaining security in the network and possibly encrypting data is also a challenge. The phenomenon that we want to study better is the reflection of waves from the stratosphere. We will also be conducting tests with new mobile scintillation detectors in the near future, which will be immediately integrated with transmitting stations.

### 6.5. Detection Efficiency

Independently of the available cosmic-ray infrastructures and expertise that is or might be contributing to the implementation of the CREDO strategies, the novel hardware extensions of the global network of detectors require new studies and tools providing information of detection efficiency on different identification and reconstruction levels: a) individual particles and the corresponding detection rates, b) extensive air showers and the corresponding cosmic-ray fluxes, and finally c) Cosmic-Ray Ensembles. The level a) is being addressed elsewhere within this Special Issue [7], considerations of level c) were initiated with Ref. [198], and a study dedicated to arrays of portable detectors such as CREDO-Maze described above is under preparation.

## 7. Data Management and Analysis

### 7.1. CREDO IT Infrastructure

From its very beginning the CREDO initiative assumed a necessity for a scalable data acquisition and processing infrastructure. Although most of the infrastructure is supplied by volunteers, in the form of smartphones and other detectors, there still exists a need to create a central repository of detection events. Information about each individual detection event is recorded first by the end device, then it is transferred to the central repository. This repository allows for managing and sharing information among interested parties, it serves as a base for CREDO data analysis. It also fulfills the role of the system's central point, providing APIs (Application Programming Interface) and information about stored data itself. The CREDO ecosystem, with some of the implementation details, is depicted in Figure 17.

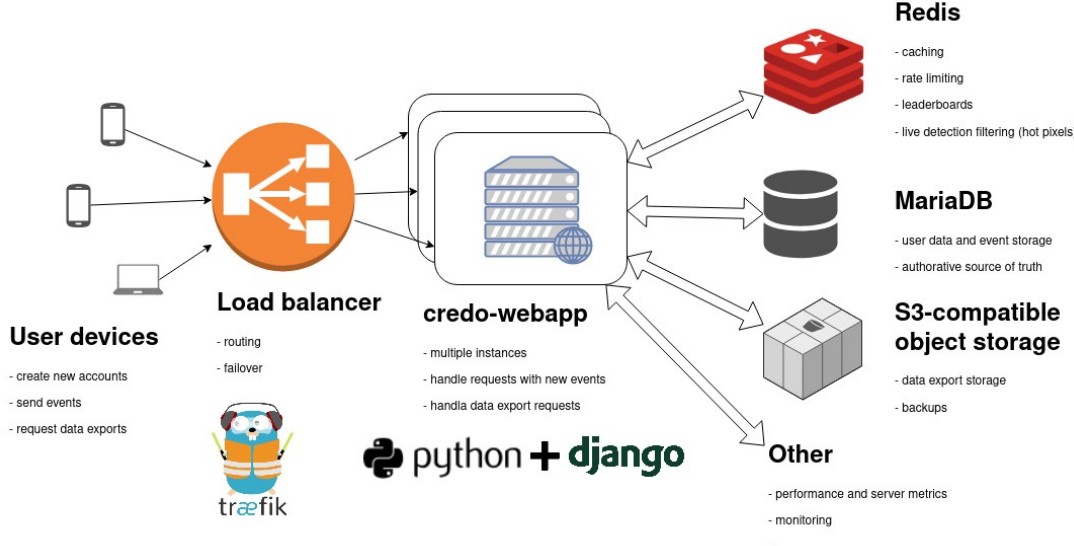

**Figure 17.** Diagram representing the architecture of storage application.

The CREDO data repository uses a dedicated API, tailored for the project's needs. The object structure used in the API closely models the information about physical detection events and additionally contains metadata, e.g., what detector was used or what was the operating time of the detector. Data storage in CREDO project is provided with respect to the FAIR principles [199], namely we provide means to Find the data, Access it, enable Interoperability and Reuse of the data. One of the means to achieve this goal is the API, which was implemented as a set of REST services adhering to the OpenAPI standard [200]. OpenAPI is a widely accepted service standard, which includes community-developed guidelines for developing interfaces exposing data and metadata. This standard focuses on data openness, technical accessibility and discoverability,

which allow for effortless integration. The stored data is publicly accessible, and so it will be in the future, which should enable the reuse of information and verification of experiments based on the given data.

Central processing for CREDO data storage was implemented as an extensible database, a storage system running in a containerized environment with aim to enhance reliability, portability and security. The main component, responsible for gathering and managing data, was implemented as a Django [201] web application. User devices communicate with the API exposed by the main application, the data is subjected to basic filtering and then placed in the storage back end. The storage is supplied through the usage of MariaDB [202] relational database cluster. Redis [203], a key-value in memory object store is used as a temporary cache, implemented to speed up queries and data presentation. The external service providing S3 compatible access is used for backups and exporting, prepared beforehand, larger chunks of data.

Due to the nature of observed physics phenomena and the community factor, data stream characteristics are often difficult to predict. Additionally, data analysis is done mainly in an exploratory manner, so the underlying hardware infrastructure needs to be flexible enough to adapt to requirements of the moment. The presented system has been deployed in a production environment (in this case, virtual machines are hosted in the cloud provided by ACC Cyfronet AGH) and is continuously gathering information about detection events from the CREDO detector network. There are several performance metrics, i.e., CPU usage and request latency, which are constantly monitored thus allowing us to determine if the current hardware configuration delivers required performance. In the case of inadequate resources, the cloud management system is able to autonomously and proactively spawn additional service instances which will allow for load distribution and handling of the data influx.

It is important to emphasize that the adopted solutions and protocols enable two-end open access: data collected by various detectors can be transmitted to the central system if only their format is kept compatible with the CREDO API structure, and also the access to the data being stored centrally is being granted to everybody upon request including a sensible motivation. The web-based monitor of the CREDO data acquisition system including basic user and detection statistics is available publicly [204] and the technical information regarding the full data access is being provided upon approving of individual requests.

*7.2. The Current Data Set*

Regardless of the ongoing efforts concerning the FAIR principles, the CREDO data set is continuously available on request for individual users if a sensible motivation for the usage is presented. In addition, all the currently available scripts and tools facilitating data access, selection, and further processing are made freely available in the public CREDO Collaboration repository [205] as a project that can be imported into the PyCharm integrated development environment [206].

The CREDO data set is divided into three basic categories:

1. Detections—a set of detections containing detailed information about individual events on all devices;
2. Pings—activity logs of devices, including the information about their connections to the database and time of work in the detecting mode;
3. Mappings—three collections containing information about users, devices and teams.

Most of the data collected by CREDO to date comes from smartphones with the CREDO Detector app, operating on the Android system. The data statistics since the premiere of the app until 1 September 2020 includes:

- 11,150 users (unique accounts) have registered
- 15,739 devices were used for particle detection
- 4,941,133 candidate detections registered
- the total operating time of the devices is over 379,629 days (over 1039 years)

The raw data files currently occupy 44 GB (detections: 39.3 GB; pings: 1.6 GB; mappings: 3.1 GB). Assuming the single CMOS camera sensor has a diagonal in size of $1/3''$ (on average), the total area of all the presently registered devices is 0.56 m$^2$. The daily average value of detections per device calculated from the number of candidate detections registered and the number of total operating time of the devices is about 13 candidate detections registered daily. Currently standard data filtering and clusterization are being performed continuously, and very early results show a perspective for more advanced image processing and classification using machine learning or deep learning techniques to perform more efficient classification of candidate detections.

A single detection record is stored in JavaScript Object Notation (JSON) open standard data format and it contains the following information:

1. user—detection user information: "team_id", "user_id";
2. location—geographical coordinates: "latitude", "longitude";
3. time—detection time information: "timestamp" (detection unix time in milliseconds), "time_received": reception time in the database;
4. picture—detection image information: "id" (unique detection identification), "frame_content": image (a fragment of the snapshot, typically containing a margin of 30 pixels around the brightest pixel position—see below) code in base64, "height": resolution "vertical" dimension, "width": resolution "horizontal" dimension;
5. server side visibility—"visible" (tells whether a detection pass through the server side filters, sensitive e.g., to repeatedly flashing pixels or incompatible versions of the applications sending the data);
6. brightest pixel position—"x", "y" (row and column number of the brightest pixel).

It often happens that a single picture taken by a smartphone in the detection mode contains more than one pixel that fulfill trigger conditions and can be classified as detections. If these pixels are located sufficiently far one from another, i.e., more than the extraction margin explained in p. 4 above, then they are considered separate detections and each of them is assigned an individual detection record. In this way clearly distinguishable particle hits collected in one shot are easily identifiable as detections with the same "timestamp". An example set of particle track candidates collected by CREDO Detector is presented in Figure 18.

The nature of the penetrating radiation measurements made by CMOS sensors assumes identification of pixels that are significantly brighter than the background. The currently used algorithms allow such identification only when the visible light does not reach the sensor, i.e., with the smartphone camera covered tightly. Intentional or unintentional uncovering of the smartphone camera might result in collecting images generated by visible light, sometimes hardly distinguishable from signal excesses induced by penetrating particles (see Figure 19 for some examples).

One might apply a specific software filter to remove such a contamination from the data sample being analyzed. The internally available preliminary studies in this direction show that the number of bright pixels that compose the image might be a sufficient data quality measure. The rule of thumb currently used within the CREDO Collaboration is based on the data collected by several "trustable" devices (operated by non-anonymous team members engaged in the application development) and it shows that one does not expect detections of penetrating particles that generate tracks composed of more than 70 pixels with brightness above 70, in the 0–255 scale.

Another type of contamination is the electronic noise which typically gives images composed of individual (or very few) bright pixels only that are being recorded very frequently compared to the expected detection frequency of the cosmic and local radiation. These electronic artifacts can be "detected" e.g., if the pixel brightness threshold is improperly set, and such a situation can be identified by monitoring the detection frequency and comparing it to the expected 100% sensor efficiency for the sensor in use. For an example, a typical CMOS sensor in a smartphone with a surface of 0.2 cm$^2$ can see at most one cosmic background muon in 5 minutes (the expected integrated background muon flux is 1/cm$^2$/minute). Assuming that the local radioactive sources might induce a signal at most

~10 times more often, one does not expect detection rates larger than 2/minute, and this is in fact the case when looking at the statistics collected from the reference ("trustable") devices. An example of the corresponding filter being used with the CREDO Collaboration requires that detection frequency is less than 10 detections with different timestamps per minute.

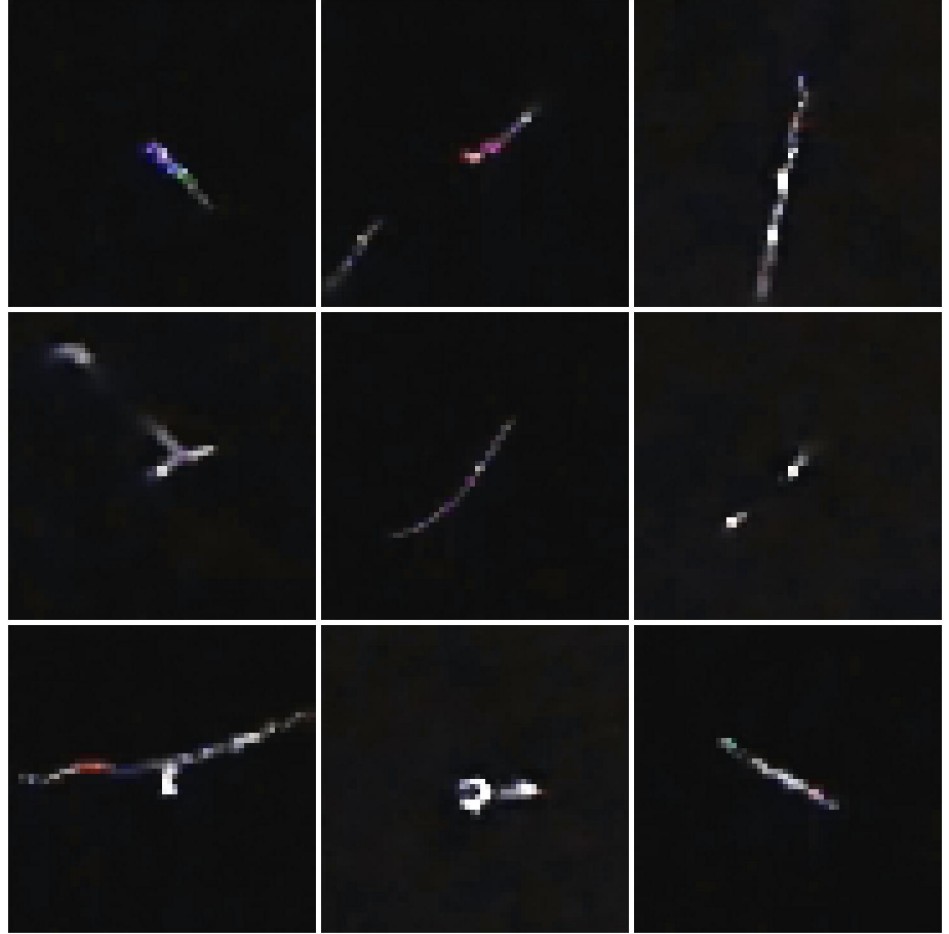

**Figure 18.** Example particle candidate tracks recorded with a smartphone using the CREDO Detector mobile application. [source: the CREDO Collaboration materials and measurements].

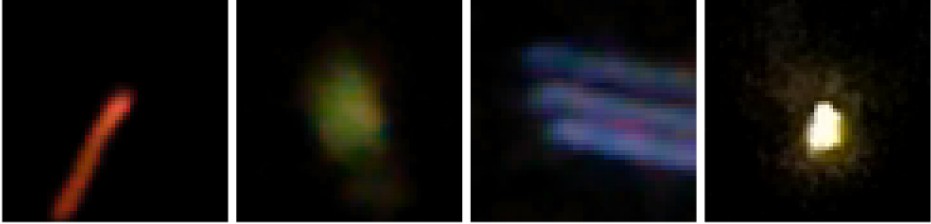

**Figure 19.** Example artifact tracks recorded with a smartphone using the CREDO Detector mobile application. [source: the CREDO Collaboration materials and measurements].

Despite the fact that with full access to the raw data sets users are able and welcome to apply their own filters and perform their own analyses, the CREDO Collaboration will periodically release its official data sets and recommendations concerning the data quality which will be based on the appropriate and publicly available studies. If it comes to scientific results, the researchers will be expected either to refer to the official data sets, or to describe and justify their own selections.

The examples of the already ongoing analyses concerning the CREDO data set include image classification based on the shapes of particle track candidates [8], monitoring detections collected by individual devices in search for temporal event clustering in 5 minute intervals [207] and muon identification and zenith angle reconstruction based on the lengths of the rectilinear particle tracks [208].

One of the examples of novel and promising directions of analyses related to the CREDO data is exploitation of the techniques of cyclostationary signal processing and its generalizations [209,210]. Cyclostationarity is a statistical property of science data generated by the combination/interaction of periodic and random phenomena. That is, these data have second- or higher-order statistical functions that are periodic functions of time. More general models can account for the presence of multiple, possibly incommensurate, and irregular periodicities ([210], Chapters 1,2). The observed signals are not periodic, but the hidden periodicity can be restored by estimating statistical functions from the data. These statistical functions contain information of the generating mechanism of the data that cannot be extracted starting from the classical stationary modeling of the observed signals.

The extensions of cyclostationarity can be of interest if time dilation effects must be accounted for ([209], Chapter 7). In particular, the effects of constant relative radial speed and/or constant relative radial acceleration between cosmic source and receiver can be suitably modeled by exploiting the generalizations of cyclostationarity. General time-warping of the source signal can be modeled by new and recently proposed signal models ([209], Chapter 6, [210], Chapters 12–14).

Source location and parameter estimation problems based on measurements taken at sensors very far apart separated are very interference tolerant if the source can be modeled as cyclostationary [211]. Pulsars have already been successfully modeled as cyclostationary sources [212].

The fraction-of-time distribution of the detected particles can exhibit periodic or almost-periodic behavior. This behavior can be demonstrated by first-order cyclostationary analysis ([210], Chapter 2).

*7.3. The Data Ontology*

The CREDO ambition concerning data quality and access policy is being gradually implemented since the first unique data of scientific value were recorded, stored in the CREDO central system and made public via web services, i.e. since the middle of 2018. With the growing data set volume and with the development and improvement of the data mining and interpretation tools, more and more effort was dedicated to maintain the compatibility of the available data with international standards. This effort is presently centered around implementing solutions concerning the data ontology.

The CREDO project is already gathering and integrating detections data originating from user's devices running the CREDO Detector app, but also from further sources, and further data, e.g., produced by processing and analysis. One of the visions of CREDO is also to make this data available for exploitation to the research community, but also to citizen scientists coming from the general public. This goal is already partly fulfilled by the CREDO data API described above. However, the current trend in open data publishing is to make the data a first-class web citizen by publishing it in semantic formats on the Linked Open Data (LOD) [213] network. (As of July 2020, the LOD network featured at least 1260 datasets from domains such as geography, life sciences, linguistics, and many others, as documented by the LOD Cloud [214].) For this, a suitable ontology (also sometimes called linked data vocabulary) needs to be used that provides semantic schema for the data. Such ontologies are usually developed in OWL (Web Ontology Language) [215] and published on the web. The data is then translated into the RDF (Resource Description Framework) [216] format using classes and properties defined in the ontology. Larger datasets are often made accessible via SPARQL endpoints—web interfaces by which users and applications may directly query the data using the SPARQL querying language [217].

The CREDO Ontology [218] is the first step towards making CREDO data available through the LOD network. The CREDO ontology is a lightweight OWL ontology. The current version features 20 classes, 14 object properties (that represent relations between classes) and 40 data properties (that represent data attributes). A simplified schema of the ontology is depicted in Figure 20.

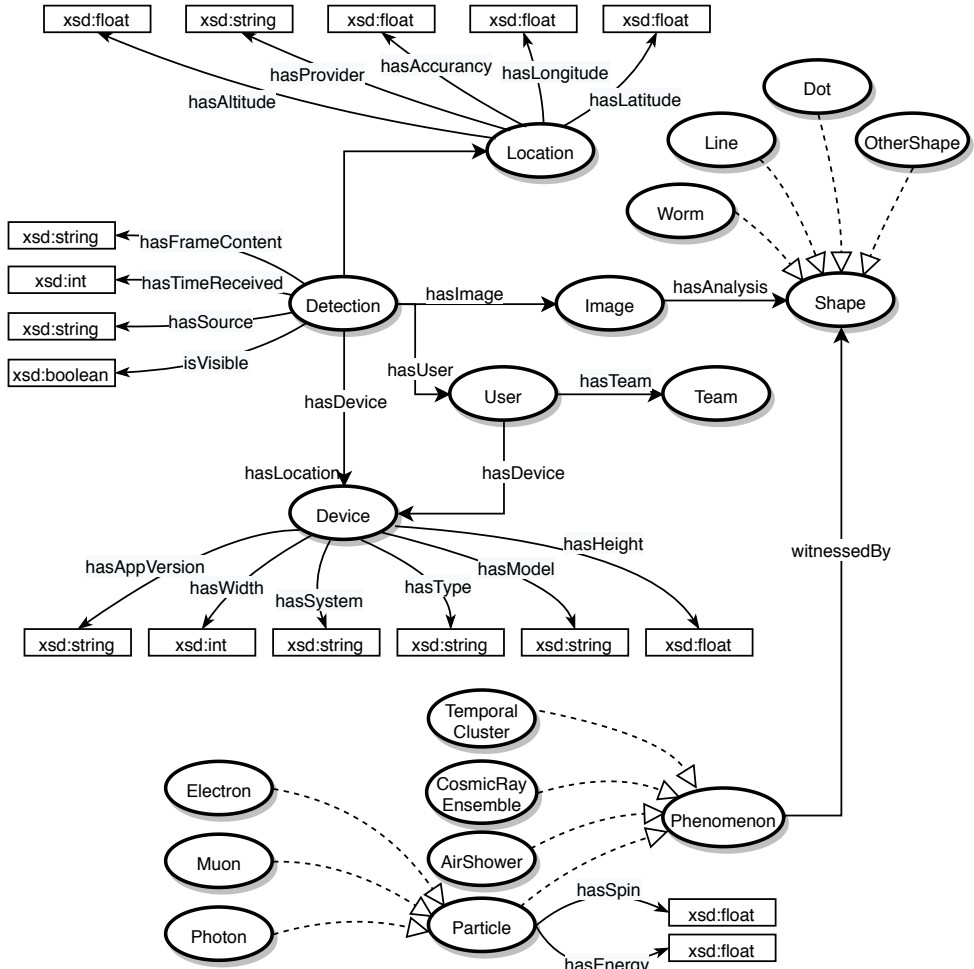

**Figure 20.** Simplified schema of the CREDO Ontology. Classes appear as ovals, data types appear as boxes. Properties of classes appear as solid arrows, while the subclass relation appears as dashed arrows with triangular heads. For legibility reasons the prefix `credo:` is omitted from classes and property names.

One part of the classes provides schema for the raw detection data that is produced by the CREDO Detector app and other sources and is currently available at the CREDO servers. Instances of the `credo:Detection` class capture data associated with each detection. Some of these are directly expressed using data properties; however, the data about the user, their device, the image captured and the location of the detection are associated as instances of the respective class `credo:User`, `credo:Device`, `credo:Location`, and `credo:Image`.

In addition, the ontology features classes that do not directly correspond to raw data recorded during detections, but rather they are intended to capture information derived from the analysis of these data. This includes the class `credo:Shape` and its four (mutually disjoint) subclasses `credo:Dot`, `credo:Line`, `credo:Worm`, and `credo:OtherShape`. These are used to classify the artifacts found in the detection images based on their geometric shape analysis.

Finally, if scientific analysis of the detected artifacts, their time frames and shapes identify that they signify a manifestation of a physical phenomenon, such phenomena can be captured under the `credo:Phenomenon` class and its subclasses, for example `credo:Particle`, `credo:AirShower`, `credo:TemporalCluster`, and others.

Although the CREDO data is currently available in a non-semantic format via the CREDO API as described above, the full utility of the CREDO ontology will be reached once the data is also available in the semantic format via a SPARQL endpoint. This is part of our ongoing efforts.

## 8. Building the Scale: Public Engagement as a Scientific Need

As explained in more detail in Ref. [3], a global scale and massive, geographically spread data acquisition effort with as many detectors as possible, even small CMOS sensors in smartphones, is essential for the success of the CREDO mission. Thus, to build up the project scale, the CREDO Collaboration considers public engagement in the project a methodological must: beneficial both for the CREDO mission and for those who contribute to its implementation. This approach underlies an effort to forge science experts and non-professionals as one community, working together towards common objectives.

The cosmic-ray dataset to be collected and processed by CREDO is expected to be of unprecedented size and complexity in the field. It will inevitably require continuous oversight and monitoring by many contributors. Thus, the CREDO public engagement effort must be planned, and often pioneered, on many levels, including massive data capture with smartphones, basic data monitoring, mining and analysis via the Internet, gamification, edutainment, and other activities. Offering these kinds of opportunities will help CREDO attract participants ready for long-lasting educational and developmental adventures. For many this will build upon the excitement of their first, up close and active involvement in cutting-edge science, such as after their first detections of particle track candidates with their smartphones. The scale of the CREDO network is planned to be as large as possible to increase the chances of scientific success. Importantly, such a strategy is also expected to bring benefits to both the science community and society. The intrinsic inclusiveness of the CREDO participation format, based on its universally appreciated educational opportunities provided in tandem with instant and open access to a premier science mission for often excluded countries, regions and individuals, has the potential to inspire and drive development to wider society on a global scale.

One of the basic (very simple) trigger concepts related to CRE and public engagement potential concerns global cosmic-ray patterns, is illustrated in Figure 10, using an example toy CRE simulation (arbitrarily dense and wide particle front arriving to the Earth from geographical North) and some random noise. The simulations are thrown on top of a global network of cosmic-ray detectors which, for simplicity, detect all the particles. One sees that a strong enough CRE signal can give a visible, global pattern built of the stations that see particles within some predefined time window. The pattern can be visualized by putting the locations of the active stations on the globe map and color-coding average arrival times of all particles hitting the detectors (top panel of Figure 21). One can also show the arrival time scatter plot (bottom-left) or/and average arrival times for the stations located within a certain territory (bottom-right—here within the 10 degree wide latitudinal belts). A random pattern is expected to give the average values located, within the uncertainties, in the middle of the considered time window, while the CRE-like signal should be visible as a departure from the "randomness line". So the pattern classification is based on a distinction between "flat line in the middle of the window" and "deviations from the flat line in the middle". This fundamental simplicity is then similar e.g., to the popular citizen science project designed to search for new planets, Planet Hunters [219], where the light curves of stars are exposed to a crowd-sourced classification and the users are asked to look for "transits" visible as periodic reduction of the received star light. This kind of pattern recognition is of course easily performed with simple algorithms or neural networks, nevertheless humans are still needed to classify the images "on the edge" or "super-strange"—seen for the first time and thus obviously not included in the training sets. In Planet Hunters the number of images suitable for human-based classification is large, and one expects even larger data load in the CREDO citizen science interfaces dedicated to global patterns: Private Particle Detective (still in the prototype phase) Dark Universe Welcome [220], both powered by Zooniverse [221], the platform which also hosts Planet Hunters. Thus, in both cases the involvement of a large number of users is not just a funny outreach option, it is a must of fundamental importance: there might be only a few really important observations (the most Earth-like planets in the case of Planet Hunters or the clearly non-random super-preshowers

in the case of Dark Universe Welcome) and we do not want to miss them. All hands (smartphones, other detectors) are needed on board.

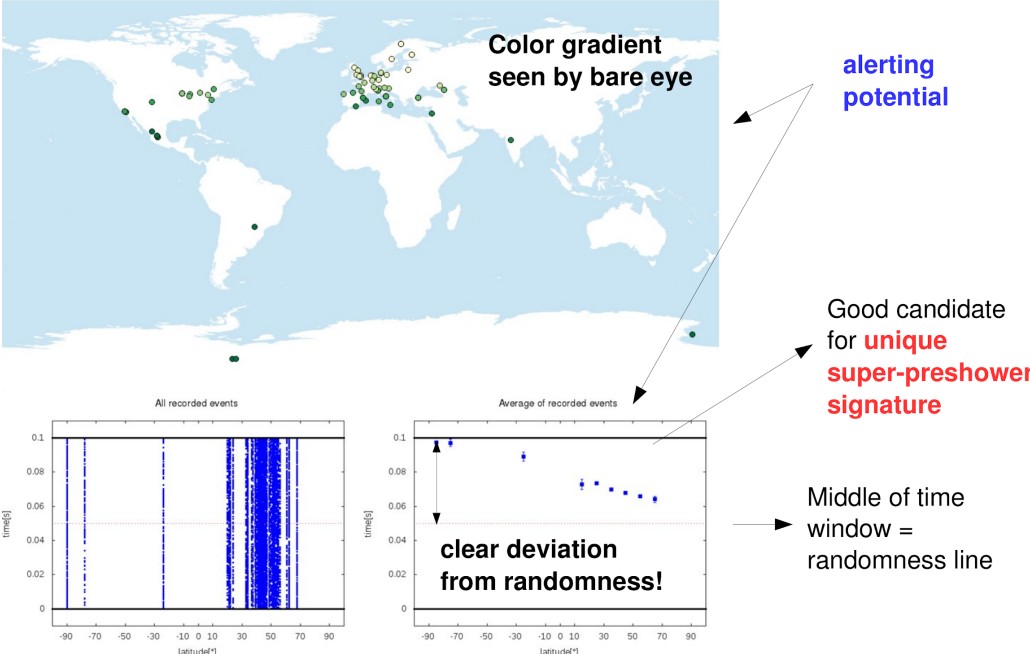

**Figure 21.** The concept of the CRE global pattern detection/alerting. See the text for details [3].

Apart from the central public engagement effort there is and will be several peripheral and potentially beneficial activities initiated by both professionals and non-expert participants of the CREDO program. The room for having and developing one's own initiatives and ideas that can be potentially available and useful to the whole CREDO community has already been proven to serve as an efficient public attractor. Example results of non-expert peripheral activities related to CREDO include e.g., Windows/Linux applications using cameras as radiation detectors [222,223], or a set of Python and SQL tools for basic data visualization and analysis [224,225] (see an example screenshot in Figure 22).

A very promising direction, both in terms of data acquisition and public engagement, is a detector application that can be run using a web browser independently of the operating system of the device. The project named "CREDO-web-detector" is now being developed by the students of Cracow University of Technology. It is an attempt to recreate the Android version of the CREDO Detector mobile application in a web browser environment. Web browsers offer a good opportunity to reach new participants, as their implementations are present in all popular operating systems. If a web-browser-based detector application is used in future, CREDO can be universally opened to Microsoft Windows, Apple IOS or Linux users. The trial version of CREDO-web-detector application is available in the public CREDO software repository [226].

Another way of public engagement stimulation and improving the user retention rate that has already been implemented in CREDO is based on introducing gamification and edutainment components into scientific projects. For example, smartphone data acquisition and particle identification studies are noticeably supported by the prototype team competition in particle detection named Particle Hunter League [227]. During the initial phase of operation advertised mostly among teachers in southern Poland the competition attracted around 2000 pupils from ∼100 teams (schools), and it is now being planned to internationalize the format beginning from the 17 CREDO member countries.

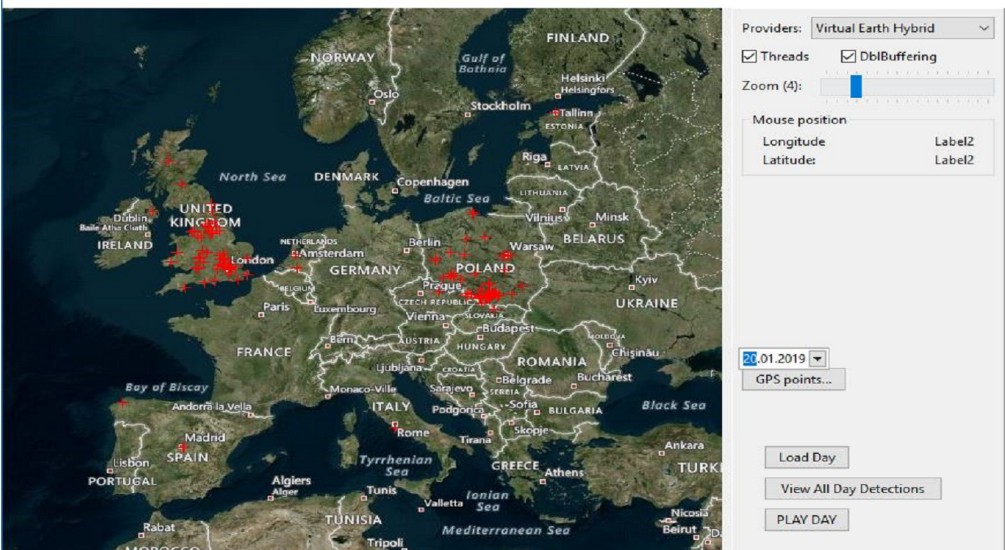

**Figure 22.** A basic visualization of the geographical locations of CREDO detections performed by a script developed by one of the non-expert CREDO participants.

Apart from a set of easy access solutions CREDO offers also higher-level opportunities such as virtual summer practices, supervision of bachelor, master and PhD projects, and also openness of the scientific teams dedicated to specific tasks reflecting the central CREDO agenda. More advanced CREDO participants and users oriented mostly on software development can also benefit from the fact that CREDO operates using open source codes under the MIT license [228]. This kind of openness enables healthy competition among the teams of software developers in a hackathon-like style, leading hopefully to a "natural" evolution of the software solutions within the CREDO program, although to make use of such non-central software activities a regular quality and functionality assessment will have to be implemented centrally, with redirecting the most promising developers and their work to the further stages, with larger scale funding.

The above highlights and examples well illustrate the societal ambition of CREDO which can directly be inferred from the deep and novel sense of the scientific research described above, where the active and massive participation of the public is rather a necessity than an option. Based on how the first CREDO users reacted to our concept, and how hungry they were to participate in a deeply important scientific project as meaningful members – we are ambitious to ignite a new era in science based on a deep public engagement in conducting a scientific research, by using smartphones in the quest for understanding the Universe to the deepest, fundamental level, hence bringing the worlds of science professionals and "just" enthusiasts closer to each other than ever.

## 9. Outlook

The current status of the Cosmic-Ray Extremely Distributed Observatory described above as well as the dynamics of the progress being made by the CREDO Collaboration allows an optimistic outlook regarding the future perspective of further development and the resulting accomplishment of the ambitious scientific objectives. However, the apparently bright and scientifically exciting future of the CREDO mission still requires attention and care today. Apart from the already functional, fast growing own original data acquisition system which provides first scientific results the CREDO concept needs to attract more intensely the wide astroparticle physics community to make an optimum use of the available infrastructures, data reservoirs and expertise resources—as must be required by any global approach to the observation and studies on cosmic-ray ensembles. This need naturally does not imply any reorientation or renaming of the existing scientific collaborations and programs, it just defines a chance for synergy leading to new observations in the field of astroparticle physics within an unprecedentedly wide energy regime, including the highest energies known. One of

the pillars of the awaited success is tightly connected with the principles of Open Science [229] focused on making scientific research (including publications, data, physical samples, and software) and its dissemination accessible to all levels of an inquiring society. One of the advantages of the Open Science research is a potential contained in the inter-collaborative effort dedicated to multi-messenger and multi-mission projects, reflected e.g., in the recent NASA initiative named Time Domain Astronomy Coordination Hub (TACH) [230], see Figure 23. CREDO directly addresses the TACH perspective, although the possible inter-cooperative activities might have different faces. One of the example directions that have been already explored is based on the expectations the gamma rays collected by CTA might be components of Cosmic-Ray Ensembles that CREDO has argued in the aforementioned Ref. [153], and there might me more direct links to the other ESFRI [231] infrastructures. For example, the unwanted muon background received by KM3NeT [232] (or any other underwater neutrino observatory!) might serve as a signal for the CRE-oriented strategies, and E-ELT [233] (and other telescopes as well!) can contribute to CREDO by sharing their unwanted cosmic-ray "artifacts" recorded with the CCD cameras in the "dark frame" mode or during regular operation. In addition, these are only selected examples given to illustrate the "connectivity" of the CRE research. We indicate places where the planned CREDO multi-messenger and multi-mission contributions can be made on top of the NASA perspective in Figure 23: (1) high-energy cosmic rays and cosmic-ray ensemble alerts provided by CREDO can naturally contribute to TACH via the Astrophysical Multi-messenger Observatory Network (AMON) [234] dedicated to high-energy astroparticle physics phenomena, and (2) the alerts on transient anomalies in secondary cosmic-ray radiation detection rate can be sent directly to the Multi-mission Transient Database. CREDO will of course benefit also from receiving alerts from TACH which will expand and enrich the scope of scientific research opportunities (enabling e.g., also studies on the sub-threshold level) available to the CREDO participants and users. The planned connection to AMON, and further to the TACH network, will give an opportunity to further enhance the credibility of the CREDO data, increase its scientific potential, and stimulate the CREDO infrastructure development adequately to the expected scientific and societal interest.

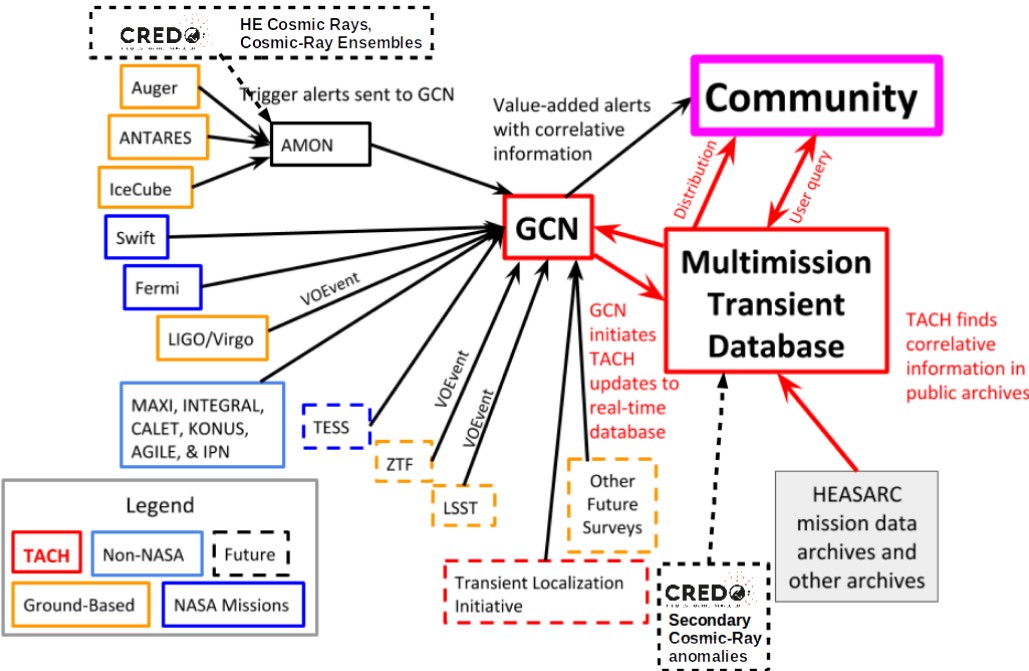

**Figure 23.** The CREDO potential contributions to the Time Domain Astronomy Coordination Hub (TACH), a new NASA initiative. The CREDO logo has been positioned in two distinct places on top of the slide by Judith Racusin, NASA, from her invited talk [235] at the New Era of Multi-Messenger Astrophysics Conference, Groningen, March 2019.

It is worthwhile stressing that the multi-mission CREDO has the potential to connect astronomy and astroparticle physics communities with other fields, creating inter- and trans- disciplinary opportunities of vital import for everyday life. Such important opportunities include e.g., studying potential correlations of transient anomalies in detection rates of low-energy secondary cosmic rays with seismic effects. The search for such correlations will be carried out based on multi-messenger signals received by global networks of detectors, including cosmic-ray sensitive devices operated by CREDO. Mass movements inside the Earth that lead to earthquakes simultaneously cause temporary changes in the gravitational and geomagnetic field. These changes are propagating at the speed of light and can therefore perhaps be observed on the surface earlier than earthquakes, for example by registering changes in the frequency of detection of cosmic radiation, which is very sensitive to geomagnetic conditions. Confirming of such correlations would give hope for a warning system, and an effort in this direction has a motivation described in the literature [236–240]; however no infrastructure suitable for global studies has been available.

Another interesting interdisciplinary line of work with CREDO is Space Weather (and Space Climate over long periods of time). Besides a lot of possibilities for specialists, CREDO can be used as an excellent introduction to space weather for the general public since cosmic rays are a fundamental part of the space environment. Until now, the classic experiment that served as an introduction to space weather for the general public was the counting of sunspots with a modest telescope [241]. However, the generalization of mobiles makes CREDO an ideal instrument to engage the general public interested in Space Weather.

Another interdisciplinary opportunity is global monitoring of the locations and arrival times of extensive air showers. Although local, and natural, radioactivity is known to transmit radiation doses significantly higher than the average secondary cosmic-ray flux, it is not yet known what impact on biological systems comes from irradiation by very energetic EAS, especially their central regions. Particle mass composition, density and energy distributions there contribute to a single radiation dose which is hard or infeasible to mimic in laboratories, and thus not available for direct reflecting/projecting in basic laboratory research. The CREDO infrastructure will enable this sort of pioneer investigation by connecting geographic coordinates and arrival times of energetic EAS cores with the locations of humans engaged in the project, providing the data for long-term studies on possible correlations between cosmic radiation and our health. Such studies will potentially lead to explaining the causes of some diseases of unknown etiologies, with hope for new therapies and/or preventive examination strategies. This interdisciplinary research will go beyond the existing research frameworks both in the fields of the physics of cosmic radiation itself and concerning the knowledge about the possible biological response of the human body to this radiation. Our studies will be conducted in accordance with the global trends in understanding the impact of low doses of radiation on living organisms in terms of the possibility of the occurrence of both positive phenomena (e.g., increasing the immunity of organisms) and negative ones (e.g., some types of cancer) [242]. These studies may shed light on interesting facts that still cannot be explained as the strong inverse correlation found in Italy between high solar activity and incidence of schizophrenia and bipolar disorder [243].

## 10. Summary and Conclusions

With CREDO we are going to explore a new observation channel of the Universe, the CRE channel. The width and complexity of the scope of the studies point to a long-term research perspective, and to the need for engaging wide communities of both professional scientists and non-expert science enthusiasts. The outcome of this program will tell whether the CRE channel broadcasts with New Physics or "just" new upper limits to astrophysical and cosmological scenarios based on null CRE results. If CRE are observed, they could point us back to the interactions at energies close to the GUT scale and lead us to a breakthrough in physics. If CRE are not observed, we will set the new upper

limits constraining selected theories, e.g., the SHDM models, which will point to a new way to narrow the search for a breakthrough in science. You are welcome to join us!

**Author Contributions:** Conceptualization, P.H., T.W.; methodology, D.B. (Dmitriy Beznosko), N.D., P.G., K.G., M.H., P.H., M.K. (Michał Karbowiak), R.K. (Renata Kierepko), S.K., M.M., J.W.M., A.N., M.N., C.O., K.O., M.P. (Maciej Pawlik), K.S. (Katarzyna Smelcerz), O.S., T.W.; software, Ł.B. (Łukasz Bibrzycki), M.B., D.B. (Dariusz Burakowski), A.Ć., N.D., A.R.D., P.G., M.K. (Marek Knap), S.K., M.M., J.M. (Justyna Mędrala), M.N., K.O., M.P. (Maciej Pawlik), M.P. (Marcin Piekarczyk), B.P., S.S., O.S.; validation, M.B., P.H., J.M. (Justyna Mędrala), B.P., S.S., O.S.; formal analysis, J.P., W.S., T.W., K.W.W.; investigation, M.B., K.A.C., N.D., P.H., J.M. (Justyna Mędrala), B.P., J.P., W.S., S.S., O.S., T.W., K.W.W.; resources, P.H., M.K. (Marek Knap), M.M., M.N., B.O., K.O., M.P. (Maciej Pawlik), M.R., J.S. (Jolanta Sulma); data curation, N.D., M.H., Z.H., O.S., S.S.; data science, Ł.B. (Łukasz Bibrzycki), M.P. (Marcin Piekarczyk), K.R. (Krzysztof Rzecki); theoretical studies, Ł.B. (Łukasz Bratek), D.E.A.-C., N.D., D.G., J.J., M.V.M., Y.J.N., J.Z.-S., A.T.; writing—original draft preparation, D.B. (Dmitriy Beznosko), Ł.B. (Łukasz Bratek), D.E.A.-C., K.A.C., N.D., P.G., K.G., D.G., A.C.G., J.J., S.K., M.H., P.H., R.K. (Renata Kierepko), M.K. (Marek Knap), M.V.M., J.W.M., A.N., C.O., K.O., M.P. (Maciej Pawlik), J.P., K.R., J.Z.-S., K.S. (Katarzyna Smelcerz), W.S., S.S., O.S., A.T., T.W., K.W.W.; writing–review and editing, D.B. (Dmitriy Beznosko), G.B., L.B. (Łukasz Bratek), N.B., K.A.C., D.E.A.-C., P.D.-o., A.R.D., K.G., A.C.G., M.H., P.H., M.K. (Marcin Kasztelan), R.K. (Robert Kamiński), P.K., B.Ł., A.M., M.V.M., J.M. (Justyna Miszczyk), V.N., Y.J.N., B.O., G.O., M.P. (Maciej Pawlik), M.R., K.R., J.Z.-S., K.S. (Katarzyna Smelcerz), K.S. (Karel Smolek), J.S. (Jarosław Stasielak), S.S., O.S., M.S., A.T., K.M.T., J.M.V., T.W.; visualization, M.K. (Marek Knap), S.S., Z.H.; supervision, Ł.B. (Łukasz Bibrzycki), A.R.D., P.H., D.G., R.K. (Robert Kamiński), R.K. (Renata Kierepko), J.W.M., M.R., K.R., S.S., J.S. (Jolanta Sulma), T.W., K.W.W.; project administration, P.H., R.K. (Robert Kamiński), M.M., S.S.; funding acquisition, M.H., R.K. (Robert Kamiński), A.T. All authors have read and agreed to the published version of the manuscript.

**Funding:** This research was partly funded by the International Visegrad grant No. 21920298, and by the National Science Center in Poland, grant No. 2018/29/B/ST2/02576. N. Budnev acknowledges partial support from The Russian Federation Ministry of Education and Science (Tunka shared core facilities—unique identifier RFMEFI59317X0005), J. Zamora-Saa acknowledges support by FONDECYT (Chile), grant No. 3180032, M.V. Medvedev acknowledges partial support via the NSF grant No. PHY-2010109, the DOE EPSCOR grant No. DE-SC0019474 and the DOE grant No. DE-SC0016368, and Martin Homola is supported by Slovak VEGA grant No. 1/0778/18. J.M. Vaquero acknowledges support from Junta de Extremadura-Fondo Social Europeo (GR18097).

**Acknowledgments:** We acknowledge the leading role in the CREDO Collaboration and the commitments to this research made by the Institute of Nuclear Physics Polish Academy of Science. This research has been supported in part by PLGrid Infrastructure and we warmly thank the staff at ACC Cyfronet AGH-UST for their always helpful supercomputing support. We thank Steven Carlip, Roger Clay, Bohdan Hnatyk, Qingdi Wang, and Henryk Wilczyński for valuable remarks and discussions.

**Conflicts of Interest:** The authors declare no conflict of interest.

## Abbreviations

The following abbreviations are frequently used in this manuscript:

| | |
|---|---|
| CR | Cosmic Ray(s) |
| CRE | Cosmic-Ray Ensemble(s) |
| CREDO | Cosmic-Ray Extremely Distributed Observatory |
| EAS | Extensive Air Shower(s) |
| UHECR | Ultra-High-Energy Cosmic Rays |

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
