# Peer review of "Cosmic-Ray Extremely Distributed Observatory"

_symmetry, doi:10.3390/sym12111835_

Round 1

Reviewer 1 Report

My main suggestion is to clarify the scientific case for cosmic ray ensembles. I do not see that case ever made in a clear way with respect to either standard model like events, or BSM events: for instance, I do not see any evidence of dark matter models that would produce a CRE.

I think the CREDO enterprise is very commendable. However, the scientific case over which it rests should be strengthened.

Author Response

Response to Reviewer 1 Comments

We would like to thank the Reviewer 1 for his/her comments. Our responses (in blue) are pasted below in between the Reviewer’s comments and suggestions (black). All changes in the paper are indicated in red.

Reviewer #1:

> My main suggestion is to clarify the scientific case for cosmic ray ensembles. 

The CREDO science case is based on the search and investigation of Cosmic Ray Ensembles (CRE), i.e. groups of cosmic rays that are spatially and temporally correlated even on a larger scale. Since the cosmic-ray research to date has mainly been focused on individual, uncorrelated cosmic ray events, the CRE-related research has the potential to enter yet uncharted astroparticle physics territory and get tuned to a new information channel about the Universe. We have described this potential in the Introduction including Figure 1 as the key illustration. In the revised version we slightly modified the abstract and lines 34-50 were rewritten and we hope it will help the readers in a better understanding of the scientific aspects of our mission. 

>I do not see that case ever made in a clear way with respect to either standard model like events, or BSM events: 

The physics scenarios capable of initiating CRE are tightly connected with particles of the highest energies known, ultra-high energy cosmic rays (UHECR), which on their ways through the Cosmos undergo various interactions resulting in generating particle cascades, parts of which might reach the Earth to give us unprecedented information about the primary processes, either within the Standard Model of particles or BSM. Selected UHECR scenarios in the context of CRE are the subject of Section 3. In the revised version we have rewritten a number of paragraphs of this section and we hope that it will improve the clarity of the overall article. The changed fragments of Section 3 include the lines: 242-251, 303-366, 412-419, the caption of Figure 3, 435-498, 524-532. Moreover, to highlight the feasibility of the CRE studies, in Section 4 “CRE simulations” we point to some specific and standard physics examples where state-of-the-art simulations of CRE give promising results in terms of the observation horizon. For example, the simulations of CRE induced by synchrotron radiation of high energy electrons interacting with the galactic magnetic fields tell us that we might have a chance to observe CRE initiated by processes that occur at least within the Galaxy. In this part the following lines were rewritten: 575-578, the caption of Figure 5, 609-629, 632-635, 648-651, which we hope will increase the clarity of Section 4. 

> for instance, I do not see any evidence of dark matter models that would produce a CRE.

We address this issue in Section 3.3. “Dark Matter as a source of UHECR”, now largely rewritten, please see lines 328-366 and 412-419. 

> I think the CREDO enterprise is very commendable. However, the scientific case over which it rests should be strengthened.

There is an ongoing effort in regard to the theoretical studies highlighted in this article and the results are being prepared or in one case already submitted for publication. We believe that the CREDO concept, which actually defines a wide and long-term science program,  will attract more and more participants and with time the range of scenarios considered for a verification with a global cosmic ray infrastructure will grow, bringing us either a new understanding of physics, or new upper limits. Both directions will have scientific value which should fuel further studies and developments of the CREDO program. We have rewritten lines  1214-1234 in Section 7. and we hope that it will increase the clarity of the description of the CREDO progress potential and scale building perspective.

Reviewer 2 Report

Summary: The paper is obviously very interesting and important for all experts interesting in experimental veryfication of contemporary concepts in fundamental physical symmetries. The paper presents a detailed presentation of a new and very instructive observational program in this field. Presentation is very clear, the links to other observational programs in the field are indicated in a user-friendly form. The expected progress is formulated straightforward and in a well-readable form.

Of course, the project will be developed quite a long time and it is very difficult to predict in advance, which particular obstacle shall be faced on this road. It is why it would be out of any sense to suggest a particular development of the present text. Minor technical defects should be however fixed.I read carefully the text and reference list to recognize several points

in the references which should be double cheched:

Ref 28

Ref 38

Ref 63

Ref 93

Ref 146

Ref 150

Ref 155

Without any doubt, the paper can be published after minor improvement of the references list.

Author Response

Response to Reviewer 2 Comments

We would like to thank the Reviewer 2 for his/her comments. Our responses (in blue) are pasted below in between the Reviewer’s comments and suggestions (black). All changes in the paper are indicated in red.

Reviewer #2:

> Summary: The paper is obviously very interesting and important for all experts interesting in experimental veryfication of contemporary concepts in fundamental physical symmetries. The paper presents a detailed presentation of a new and very instructive observational program in this field. Presentation is very clear, the links to other observational programs in the field are indicated in a user-friendly form. The expected progress is formulated straightforward and in a well-readable form. Of course, the project will be developed quite a long time and it is very difficult to predict in advance, which particular obstacle shall be faced on this road. It is why it would be out of any sense to suggest a particular development of the present text. 

We fully agree with the Reviewer and appreciate his/her very grasping of the spirit and potential of the CREDO science program. 

> Minor technical defects should be however fixed.I read carefully the text and reference list to recognize several points in the references which should be double cheched: Ref 28, Ref 38, Ref 63, Ref 93, Ref 146, Ref 150, Ref 155.

We are thankful for so careful checking of the Reference list. We have double checked the references pointed out by the Reviewer and implemented the necessary changes. Only in the last case (old Ref. 155, new number 154) we left the bibliographic details as they were - the corresponding author of the cited article uses a double last name which we would like to respect.

> Without any doubt, the paper can be published after minor improvement of the references list.

We thank the Reviewer for this clear evaluation.

Round 2

Reviewer 1 Report

The authors have satisfactorily addressed the concern I expressed in my original review; I recommend this manuscript for publication